# $f$-DM: A Multi-stage Diffusion Model via Progressive Signal Transformation

**Jiatao Gu, Shuangfei Zhai, Yizhe Zhang, Miguel Angel Bautista, Josh Susskind**
Apple
{jgu32,szhai,yizhe_zhang,mbautistamartin,jsusskind}@apple.com

## ABSTRACT

Diffusion models (DMs) have recently emerged as SoTA tools for generative modeling in various domains. Standard DMs can be viewed as an instantiation of hierarchical variational autoencoders (VAEs) where the latent variables are inferred from input-centered Gaussian distributions with fixed scales and variances. Unlike VAEs, this formulation constrains DMs from changing the latent spaces and learning abstract representations. In this work, we propose $f$-DM, a generalized family of DMs, which allows progressive signal transformation. More precisely, we extend DMs to incorporate a set of (hand-designed or learned) transformations, where the transformed input is the mean of each diffusion step. We propose a generalized formulation of DMs and derive the corresponding de-noising objective together with a modified sampling algorithm. As a demonstration, we apply $f$-DM in image generation tasks with a range of functions, including down-sampling, blurring, and learned transformations based on the encoder of pretrained VAEs. In addition, we identify the importance of adjusting the noise levels whenever the signal is sub-sampled and propose a simple rescaling recipe. $f$-DM can produce high-quality samples on standard image generation benchmarks like FFHQ, AFHQ, LSUN and ImageNet with better efficiency and semantic interpretation. Please check our videos at http://jiataogu.me/fdm/.

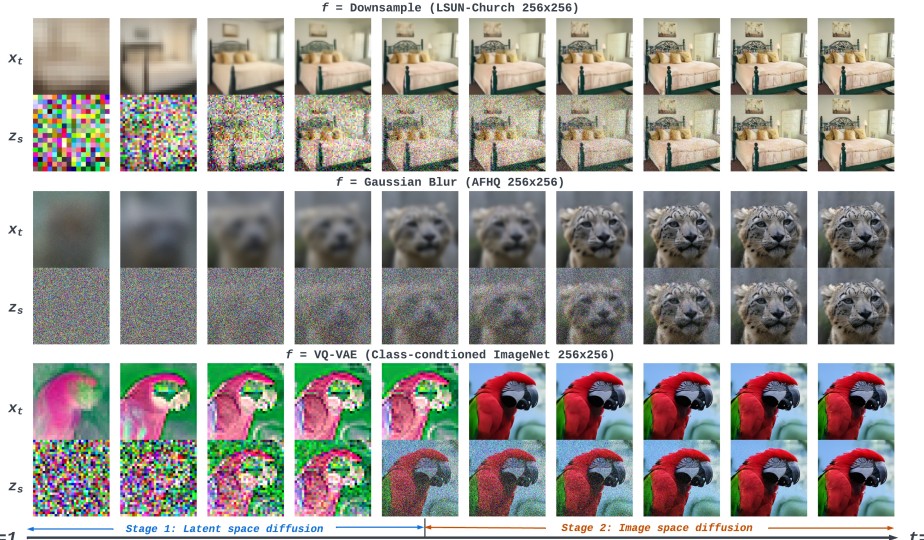

Figure 1: Visualization of reverse diffusion from $f$-DMs with various signal transformations. $x_t$ is the denoised output, and $z_s$ is the input to the next diffusion step. We plot the first three channels of VQVAE latent variables. Low-resolution images are resized to $256^2$ for ease of visualization.

## 1 INTRODUCTION

Diffusion probabilistic models (DMs, Sohl-Dickstein et al., 2015; Ho et al., 2020; Nichol & Dhariwal, 2021) and score-based (Song et al., 2021b) generative models have become increasingly popular

as the tools for high-quality image (Dhariwal & Nichol, 2021), video (Ho et al., 2022b), text-to-speech (Popov et al., 2021) and text-to-image (Rombach et al., 2021; Ramesh et al., 2022; Saharia et al., 2022a) synthesis. Despite the empirical success, conventional DMs are restricted to operate in the ambient space throughout the Gaussian noising process. On the other hand, common generative models like VAEs (Kingma & Welling, 2013) and GANs (Goodfellow et al., 2014; Karras et al., 2021) employ a coarse-to-fine process that hierarchically generates high-resolution outputs.

We are interested in combining the best of the two worlds: the expressivity of DMs and the benefit of hierarchical features. To this end, we propose $f$-DM, a generalized multi-stage framework of DMs to incorporate progressive transformations to the inputs. As an important property of our formulation, $f$-DM does not make any assumptions about the type of transformations. This makes it compatible with many possible designs, ranging from domain-specific ones to generic neural networks. In this work, we consider representative types of transformations, including down-sampling, blurring, and neural-based transformations. What these functions share in common is that they allow one to derive increasingly more global, coarse, and/or compact representations, which we believe can lead to better sampling quality as well as reduced computation.

Incorporating arbitrary transformations into DMs also brings immediate modeling challenges. For instance, certain transformations destroy the information drastically, and some might also change the dimensionality. For the former, we derive an interpolation-based formulation to smoothly bridge consecutive transformations. For the latter, we verify the importance of rescaling the noise level, and propose a *resolution-agnostic* signal-to-noise ratio (SNR) as a practical guideline for noise rescaling.

Extensive experiments are performed on image generation benchmarks, including FFHQ, AFHQ, LSUN Bed/Church and ImageNet. $f$-DMs consistently match or outperform the baseline performance, while requiring relatively less computing thanks to the progressive transformations. Furthermore, given a pre-trained $f$-DM, we can readily manipulate the learned latent space, and perform conditional generation tasks (e.g., super-resolution) without additional training.

## 2  BACKGROUND

**Diffusion Models** (DMs, Sohl-Dickstein et al., 2015; Song & Ermon, 2019; Ho et al., 2020) are deep generative models which can be viewed as a special case of hierarchical VAEs (Kingma et al., 2021). In this paper, we consider diffusion in continuous time similar to Song et al. (2021b); Kingma et al. (2021).

Given a datapoint $\boldsymbol{x} \in \mathbb{R}^N$, a DM models time-dependent latent variables $\boldsymbol{z} = \{\boldsymbol{z}_t | t \in [0,1], \boldsymbol{z}_0 = \boldsymbol{x}\}$ based on a fixed signal-noise schedule $\{\alpha_t, \sigma_t\}$:

$$q(\boldsymbol{z}_t | \boldsymbol{z}_s) = \mathcal{N}(\boldsymbol{z}_t; \alpha_{t|s}\boldsymbol{z}_s, \sigma_{t|s}^2 I),$$

where $\alpha_{t|s} = \alpha_t/\alpha_s, \sigma_{t|s}^2 = \sigma_t^2 - \alpha_{t|s}^2 \sigma_s^2, s < t$. It also defines the marginal distribution $q(\boldsymbol{z}_t | \boldsymbol{x})$ as:

$$q(\boldsymbol{z}_t | \boldsymbol{x}) = \mathcal{N}(\boldsymbol{z}_t; \alpha_t \boldsymbol{x}, \sigma_t^2 I),$$

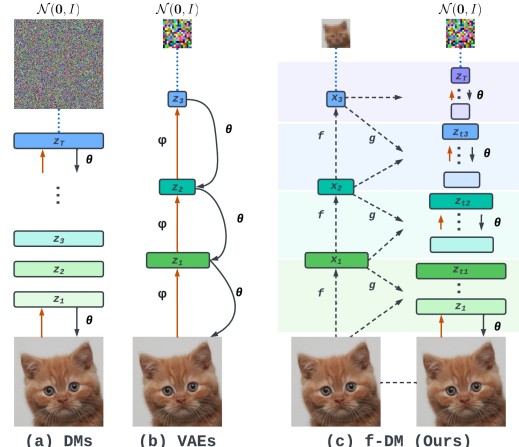

Figure 2: (a) the standard DMs; (b) a bottom-up hierarchical VAEs; (c) our proposed $f$-DM.

By default, we assume the variance preserving form (Ho et al., 2020). That is, $\alpha_t^2 + \sigma_t^2 = 1, \alpha_0 = \sigma_1 = 1$, and the signal-to-noise-ratio (SNR, $\alpha_t^2/\sigma_t^2$) decreases monotonically with $t$. For generation, a parametric function $\theta$ is optimized to reverse the diffusion process by denoising $\boldsymbol{z}_t = \alpha_t \boldsymbol{x} + \sigma_t \boldsymbol{\epsilon}$ to the clean input $\boldsymbol{x}$, with a weighted reconstruction loss $\mathcal{L}_\theta$. For example, the "simple loss" proposed in Ho et al. (2020) is equivalent to weighting residuals by $\omega_t = \alpha_t^2/\sigma_t^2$:

$$\mathcal{L}_\theta = \mathbb{E}_{\boldsymbol{z}_t \sim q(\boldsymbol{z}_t | \boldsymbol{x}), t \sim [0,1]} \left[ \omega_t \cdot \|\boldsymbol{x}_\theta(\boldsymbol{z}_t, t) - \boldsymbol{x}\|_2^2 \right]. \tag{1}$$

In practice, $\theta$ is parameterized as a U-Net (Ronneberger et al., 2015). As suggested in Ho et al. (2020), predicting the noise $\boldsymbol{\epsilon}_\theta$ empirically achieves better performance than predicting $\boldsymbol{x}_\theta$, where

$x_\theta(z_t, t) = (z_t - \sigma_t \epsilon_\theta(z_t, t))/\alpha_t$. Sampling from such a learned model can be performed from ancestral sampling (DDPM, Ho et al., 2020), or a deterministic DDIM sampler (Song et al., 2021a). Starting from $z_1 \sim \mathcal{N}(0, I)$, a sequence of timesteps $1 = t_0 > \ldots > t_N = 0$ are sampled for iterative generation, and we can readily summarize both methods for each step as follows:

$$z_s = \alpha_s \cdot x_\theta(z_t) + \sqrt{\sigma_s^2 - \eta^2 \bar{\sigma}^2} \cdot \epsilon_\theta(z_t) + \eta \bar{\sigma} \cdot \epsilon, \quad \epsilon \sim \mathcal{N}(0, I), \quad s < t, \tag{2}$$

where $\bar{\sigma} = \sigma_s \sigma_{t|s}/\sigma_t$, and $\eta$ controls the proportion of additional noise. (i.e., DDIM $\eta = 0$).

As the score function $\epsilon_\theta$ is defined in the ambient space, it is clear that all the latent variables $z$ are forced to be the same shape as the input data $x$ ($\mathbb{R}^N$). This not only leads to inefficient training, especially for steps with high noise level (Jing et al., 2022), but also makes DMs hard to learn abstract and semantically meaningful latent space as pointed out by Preechakul et al. (2022).

## 3 METHOD

In this section, we introduce $f$-DM, an extended family of DMs to enable diffusion on transformed signals, in a way similar to a standard hierarchical VAE. We start by introducing the definition of the proposed multi-stage formulation with general signal transformations, followed by modified training and generation algorithms (Section 3.1). Then, we specifically apply $f$-DM with three categories of transformations (Section 3.2).

### 3.1 MULTI-STAGE DIFFUSION

**Signal Transformations** We consider a sequence of deterministic functions $\boldsymbol{f} = \{f_0, \ldots, f_K\}$, where $f_0 \ldots f_k$ progressively transforms the input signal $x \in \mathbb{R}^N$ into $x^k = f_{0:k}(x) \in \mathbb{R}^{M_k}$. We assume $x^0 = f_0(x) = x$. In principle, $\boldsymbol{f}$ can be any function. In this work, we focus on transformations that gradually destroy the information contained in $x$ (e.g., down-sampling), leading towards more compact representations. Without loss of generality, we assume $M_0 \geq M_1 \geq \ldots \geq M_K$. A sequence of *inverse* mappings $\boldsymbol{g} = \{g_0, \ldots, g_{K-1}\}$ is used to connect a corresponding sequence of pairs of consecutive spaces. Specifically, we define $\hat{x}^k$ as:

$$\hat{x}^k := \begin{cases} g_k\left(f_{k+1}(x^k)\right) \approx x^k, & \text{if } k < K, \\ x^k, & \text{if } k = K. \end{cases} \tag{3}$$

The approximation of Equation 3 ($k < K$) is not necessarily (and sometimes impossibly) accurate. For instance, $f_k$ downsamples an input image $x$ from $128^2$ into $64^2$ with average pooling, and $g_k$ can be a bilinear interpolation that upsamples back to $128^2$, which is a lossy reconstruction.

The definition of $\boldsymbol{f}$ and $\boldsymbol{g}$ can be seen as a direct analogy of the encoder ($\phi$) and decoder ($\theta$) in hierarchical VAEs (see Figure 2 (b)). However, there are still major differences: (1) the VAE encoder/decoder is stochastic, and the encoder's outputs are regularized by the prior. In contrast, $\boldsymbol{f}$ and $\boldsymbol{g}$ are deterministic, and the encoder output $x^K$ does not necessarily follow a simple prior; (2) VAEs directly use the decoder for generation, while $\boldsymbol{f}, \boldsymbol{g}$ are fused in the diffusion steps of $f$-DM.

**Forward Diffusion** We extend the continuous-time DMs for signal transformations. We split the diffusion time $0 \rightarrow 1$ into $K+1$ stages, where for each stage, a partial diffusion process is performed. More specifically, we define a set of time boundaries $0 = \tau_0 < \tau_1 < \ldots < \tau_K < \tau_{K+1} = 1$, and for $t \in [0, 1]$, the latent $z_t$ has the following marginal probability:

$$q(z_t|x) = \mathcal{N}(z_t; \alpha_t x_t, \sigma_t^2 I), \quad \text{where } x_t = \frac{(t - \tau_k)\hat{x}^k + (\tau_{k+1} - t)x^k}{\tau_{k+1} - \tau_k}, \quad \tau_k \leq t < \tau_{k+1}. \tag{4}$$

As listed above, $x_t$ is the interpolation of $x^k$ and its approximation $\hat{x}^k$ when $t$ falls in stage $k$. A simple illustration for the relationship of $x_t, \hat{x}^k, x^k$ and $z_t$ is shown in Figure 10. We argue that interpolation is crucial as it creates a *continuous* transformation that slowly corrupts information inside each stage. In this way, such change can be easily reversed by our model. Also, it is non-trivial to find the optimal stage schedule $\tau_k$ for each model as it highly depends on how much the information is destroyed in each stage $f_k$. In this work, we tested two heuristics: (1) *linear* schedule $\tau_k = k/(K+1)$; (2) *cosine* schedule $\tau_k = \cos(1 - k/(K+1))$. Note that the standard DMs can be seen as a special case of our $f$-DM when there is only one stage ($K = 0$).

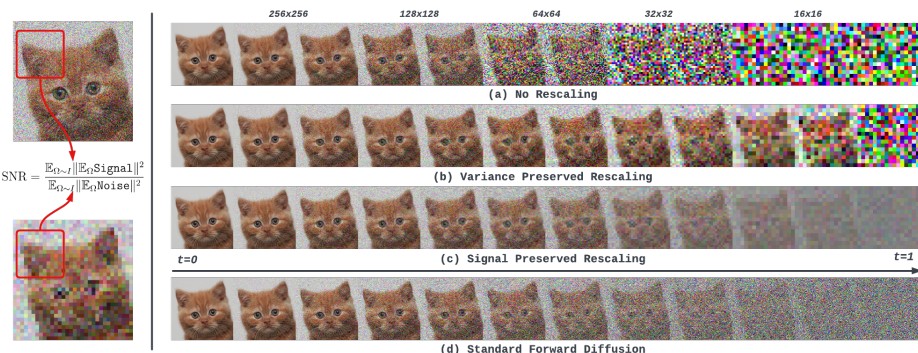

Figure 3: Left: an illustration of the proposed SNR computation for different sampling rates; Right: the comparison of rescaling the noise level for progressive down-sampling. Without noise rescaling, the diffused images in low-resolution quickly become too noisy to distinguish the underline signal.

Equation 4 does not guarantee a Markovian transition. Nevertheless, our formulation only need $q(\boldsymbol{z}_t|\boldsymbol{z}_s, \boldsymbol{x})$, which has the following simple form focusing on diffusion steps within a stage:

$$q(\boldsymbol{z}_t|\boldsymbol{z}_s, \boldsymbol{x}) = \mathcal{N}\left(\boldsymbol{z}_t; \alpha_{t|s}\boldsymbol{z}_s + \alpha_t \cdot (\boldsymbol{x}_t - \boldsymbol{x}_s), \sigma^2_{t|s}I\right), \quad \tau_k \leq s < t < \tau_{k+1}. \tag{5}$$

From Equation 5, we further re-write $\boldsymbol{x}_t - \boldsymbol{x}_s = -\boldsymbol{\delta}_t \cdot (t - s)/(t - \tau_k)$, where $\boldsymbol{\delta}_t = \boldsymbol{x}^k - \boldsymbol{x}_t$ is the signal degradation. Equation 5 also indicates that the reverse diffusion distribution $q(\boldsymbol{z}_s|\boldsymbol{z}_t, \boldsymbol{x}) \propto q(\boldsymbol{z}_t|\boldsymbol{z}_s, \boldsymbol{x})q(\boldsymbol{z}_s|\boldsymbol{x})$ can be written as the function of $\boldsymbol{x}_t$ and $\boldsymbol{\delta}_t$ which will be our learning objectives.

**Boundary Condition** To enable diffusion across stages, we need the transition at stage boundaries $\tau_k$. More specifically, when the step approaches the boundary $\tau^-$ (the left limit of $\tau$), the transition $q(\boldsymbol{z}_\tau|\boldsymbol{z}_{\tau^-}, \boldsymbol{x})$ should be as deterministic (ideally invertible) & smooth as possible to minimize information loss.[1] First, we can easily expand $\boldsymbol{z}_\tau$ and $\boldsymbol{z}_{\tau^-}$ as the signal and noise combination:

$$\begin{aligned} \textit{Before: } \boldsymbol{z}_{\tau^-} &= \alpha_{\tau^-} \cdot \boldsymbol{x}_{\tau^-} + \sigma_{\tau^-} \cdot \boldsymbol{\epsilon}, \quad p(\boldsymbol{\epsilon}) = \mathcal{N}(\mathbf{0}, I), \\ \textit{After: } \boldsymbol{z}_\tau &= \alpha_\tau \cdot \boldsymbol{x}_\tau \quad + \sigma_\tau \cdot \zeta(\boldsymbol{\epsilon}), \quad p(\zeta(\boldsymbol{\epsilon})) = \mathcal{N}(\mathbf{0}, I). \end{aligned} \tag{6}$$

Based on definition, $\boldsymbol{x}_{\tau^-} = \hat{\boldsymbol{x}}^{k-1} = g(\boldsymbol{x}^k) = g(\boldsymbol{x}_\tau)$, which means the signal part is invertible. Therefore we only need to find $\zeta$. Under the initial assumption of $M_k \leq M_{k-1}$, this can be achieved easily by dropping elements from $\boldsymbol{\epsilon}$. Take down-sampling ($M_{k-1} = 4M_k$) as an example. We can directly drop 3 out of every $2 \times 2$ values from $\boldsymbol{\epsilon}$. More details are included in Appendix A.4.

The second requirement of a smooth transition is not as straightforward as it looks, which asks the "noisiness" of latents $\boldsymbol{z}$ to remain unchanged across the boundary. We argue that the conventional measure – the signal-to-noise-ratio (SNR) – in DM literature is not compatible with resolution change as it averages the signal/noise power element-wise. In this work, we propose a generalized *resolution-agnostic* SNR by viewing data as points sampled from a continuous field:

$$\text{SNR}(\boldsymbol{z}) = \frac{\mathbb{E}_{\Omega \sim I}\|\mathbb{E}_{i \sim \Omega}\text{SIGNAL}(\boldsymbol{z}_i)\|^2}{\mathbb{E}_{\Omega \sim I}\|\mathbb{E}_{i \sim \Omega}\text{NOISE}(\boldsymbol{z}_i)\|^2}, \tag{7}$$

where $I$ is the data range, SIGNAL represents the real data value (such as image pixels), and NOISE is the unstructed Gaussian noise added to the data. $\Omega$ is a patch relative to $I$, which can be any size as long as it is invariant to different sampling rates (resolutions). As shown in Figure 3 (left), we can obtain a reliable measure of noisiness by averaging the signal/noise inside patches. We derive $\alpha_\tau, \sigma_\tau$ from $\alpha_{\tau^-}, \sigma_{\tau^-}$ for any transformations by forcing $\text{SNR}(\boldsymbol{z}_\tau) = \text{SNR}(\boldsymbol{z}_{\tau^-})$ under this new definition. Specifically, if dimensionality change is solely caused by the change of sampling rate (e.g., down-sampling, average RGB channels, deconvolution), we can get the following relation:

$$\alpha_\tau^2/\sigma_\tau^2 = d_k \cdot \gamma_k \cdot \alpha_{\tau^-}^2/\sigma_{\tau^-}^2, \tag{8}$$

where $d_k = M_{k-1}/M_k$ is the total dimension change, and $\gamma_k = \mathbb{E}\|\hat{\boldsymbol{x}}^{k-1}\|^2/\mathbb{E}\|\boldsymbol{x}^k\|^2$ is the change of signal power. For example, we have $d_k = 4, \gamma_k \approx 1$ for down-sampling. Following Equation 8, the straightforward rule is to rescale the magnitude of the noise, and keep the signal part unchanged:

---

[1]For simplicity, we omit the subscript $k$ for $\tau_k$ in the following paragraphs.

---

**Algorithm 1:** Reverse diffusion for image generation using $f$-DM

---

**Input:** model $\theta$, $\boldsymbol{f}$, $\boldsymbol{g}$, stage schedule $\{\tau_0, \ldots, \tau_K\}$, **rescaled** noise schedule functions $\alpha(.), \sigma(.)$, step-size $\Delta t$, $\boldsymbol{\epsilon}_{\text{full}} \sim \mathcal{N}(\mathbf{0}, I)$, DDPM ratio $\eta$

1  **Initialize** $\boldsymbol{z}$ from $\boldsymbol{\epsilon}_{\text{full}}$
2  **for** $(k = K; k \geq 0; k = k - 1)$ **do**
3     **for** $(t = \tau_{k+1}; t > \tau_k; t = t - \Delta t, s = t - \Delta t)$ **do**
4       $\boldsymbol{\epsilon}_\theta, \boldsymbol{\delta}_\theta = \theta(\boldsymbol{z}, t); \quad \boldsymbol{x}_\theta = (\boldsymbol{z} - \sigma(t) \cdot \boldsymbol{\epsilon}_\theta)/\alpha(t);$
5       **if** $s > \tau_k$ **then**
6          $\boldsymbol{z} = \alpha(s) \cdot (\boldsymbol{x}_\theta + \boldsymbol{\delta}_\theta \cdot (t-s)/(t-\tau_k)) + \sqrt{\sigma^2(s) - \eta^2\bar{\sigma}^2} \cdot \boldsymbol{\epsilon}_\theta + \eta\bar{\sigma} \cdot \boldsymbol{\epsilon}, \boldsymbol{\epsilon} \sim \mathcal{N}(\mathbf{0}, I)$
7     **if** $k > 0$ **then**
8       **Re-sample** noise $\boldsymbol{\epsilon}_{\text{rs}}$ from $\boldsymbol{\epsilon}_\theta$ and $\boldsymbol{\epsilon}_{\text{full}};\quad \boldsymbol{z} = \alpha(\tau_k) \cdot g_k(\boldsymbol{x}_\theta) + \sigma(\tau_k) \cdot \boldsymbol{\epsilon}_{\text{rs}}$

9  **return** $\boldsymbol{x}_\theta$

---

$\alpha \leftarrow \alpha, \sigma \leftarrow \sigma/\sqrt{d_k}$, which we refer as signal preserved (SP) rescaling. Note that, to ensure the noise schedule is continuous over time and close to the original schedule, such rescaling is applied to the noises of the entire stage, and will be accumulated when multiple transformations are used. As the comparison shown in Figure 3, the resulting images are visually closer to the standard DM. However, the variance of $\boldsymbol{z}_t$ becomes very small, especially when $t \rightarrow 1$, which might be hard for the neural networks to distinguish. Therefore, we propose the variance preserved (VP) alternative to further normalize the rescaled $\alpha, \sigma$ so that $\alpha^2 + \sigma^2 = 1$. We show the visualization in Figure 3 (b).

**Training** We train a neural network $\theta$ to denoise. We also show the training pipeline in Figure 10. In $f$-DM, noise is caused by two factors: (1) the perturbation $\boldsymbol{\epsilon}$ from noise injection; (2) the degradation $\boldsymbol{\delta}$ due to signal transformation. Thus, we propose to predict $\boldsymbol{x}_\theta$ and $\boldsymbol{\delta}_\theta$ jointly, which simultaneously remove both noises from $\boldsymbol{z}_t$ with a "double reconstruction" loss:

$$\mathcal{L}_\theta = \mathbb{E}_{\boldsymbol{z}_t \sim q(\boldsymbol{z}_t|\boldsymbol{x}), t \sim [0,1]} \left[ \omega_t \cdot \left( \|\boldsymbol{x}_\theta(\boldsymbol{z}_t, t) - \boldsymbol{x}_t\|_2^2 + \|\boldsymbol{\delta}_\theta(\boldsymbol{z}_t, t) - \boldsymbol{\delta}_t\|_2^2 \right) \right], \tag{9}$$

where the denoised output is $\boldsymbol{x}_\theta(\boldsymbol{z}_t, t) + \boldsymbol{\delta}_\theta(\boldsymbol{z}_t, t)$. Unlike standard DMs, the denoising goals are the transformed signals of each stage rather than the final real images, which are generally simpler targets to recover. The same as standard DMs, we also choose to predict $\boldsymbol{\epsilon}_\theta$, and compute $\boldsymbol{x}_\theta = (\boldsymbol{z}_t - \sigma_t \boldsymbol{\epsilon}_\theta)/\alpha_t$. We adopt the same U-Net architecture for all stages, where input $\boldsymbol{z}_t$ will be directed to the corresponding inner layer based on spatial resolutions (see Appendix Figure 11 for details).

**Unconditional Generation** We present the generation steps in Algorithm 1, where $\boldsymbol{x}_t$ and $\boldsymbol{\delta}_t$ are replaced by model's predictions $\boldsymbol{x}_\theta, \boldsymbol{\delta}_\theta$. Thanks to the interpolation formulation (Equation 4), generation is independent of the transformations $\boldsymbol{f}$. Only the inverse mappings $\boldsymbol{g}$ – which might be simple and easy to compute – is needed to map the signals at boundaries. This brings flexibility and efficiency to learning complex or even test-time inaccessible transformations. In addition, Algorithm 1 includes a "noise-resampling step" for each stage boundary, which is the reverse process for $\zeta(\boldsymbol{\epsilon})$ in Equation 6. While $\zeta$ is deterministic, the reverse process needs additional randomness. For instance, if $\zeta$ drops elements in the forward process, then the reverse step should inject standard Gaussian noise back to the dropped locations. Because we assume $M_0 \geq \ldots \geq M_K$, we propose to sample a full-size noise $\boldsymbol{\epsilon}_{\text{full}}$ before generation, and gradually adding subsets of $\boldsymbol{\epsilon}_{\text{full}}$ to each stage. Thus, $\boldsymbol{\epsilon}_{\text{full}}$ encodes multi-scale information similar to RealNVP (Dinh et al., 2016).

**Conditional Generation** Given an unconditional $f$-DM, we can do conditional generation by replacing the denoised output $\boldsymbol{x}_\theta$ with any condition $\boldsymbol{x}_c$ at a suitable time ($T$), and starting diffusion from $T$. For example, suppose $\boldsymbol{f}$ is `downsample`, and $\boldsymbol{x}_c$ is a low-resolution image, $f$-DM enables super-resolution (SR) without additional training. To achieve that, it is critical to initialize $\boldsymbol{z}_T$, which implicitly asks $\boldsymbol{z}_T \approx \alpha_T \boldsymbol{x}_c + \sigma_T \boldsymbol{\epsilon}_\theta(\boldsymbol{z}_T)$. In practice, we choose $T$ to be the corresponding stage boundary, and initialize $\boldsymbol{z}$ by adding random noise $\sigma_T \boldsymbol{\epsilon}$ to $\alpha_T \boldsymbol{x}_c$. A *gradient-based* method is used to iteratively update $\boldsymbol{z}_T \leftarrow \boldsymbol{z}_T - \lambda \nabla_{\boldsymbol{z}_T} \|\boldsymbol{x}_\theta(\boldsymbol{z}_T) - \boldsymbol{x}_c\|_2^2$ for a few steps before the diffusion starts.

### 3.2 Applications on Various Transformations

With the definition in Section 3.1, next we show $f$-DM applied with different transformations. In this paper, we consider the following three categories of transformations.

**Downsampling.** As the motivating example in Section 3.1, we let $\boldsymbol{f}$ a sequence of `downsampe` operations that transforms a given image (e.g., $256^2$) progressively down to $16^2$, where each $f_k(.)$

reduces the length by 2, and correspondingly $g_k(.)$ upsamples by 2. Thus, the generation starts from a low-resolution noise and progressively performs super-resolution. We denote the model as $f$-DM-DS, where $d_k = 4, \gamma_k = 1$ in Equation 8 and $K = 4$ for $256^2$ images.

**Blurring.** $f$-DM also supports general `blur` transformations. Unlike recent works (Rissanen et al., 2022; Hoogeboom & Salimans, 2022) that focuses on continuous-time blur (heat dissipation), Equation 4 can be seen as an instantiation of progressive blur function if we treat $\hat{x}^k$ as a blurred version of $x^k$. This design brings more flexibility in choosing any kind of blurring functions, and using the blurred versions as stages. In this paper, we experiment with two types of blurring functions. (1) $f$-DM-Blur-U: utilizing the same downsample operators as $f$-DM-DS, while always up-sampling the images back to the original sizes; (2) $f$-DM-Blur-G: applying standard Gaussian blurring kernels following Rissanen et al. (2022). In both cases, we use $g_k(x) = x$. As the dimension is not changed, no rescaling and noise resampling is required.

**Image $\rightarrow$ Latent Trans.** We further consider diffusion with learned non-linear transformations such as VAEs (see Figure 2 (b), $f$: VAE encoder, $g$: VAE decoder). By inverting such an encoding process, we are able to generate data from low-dimensional latent space similar to Rombach et al. (LDM, 2021). As a major difference, LDM operates only on the latent variables, while $f$-DM learns diffusion in the latent and image spaces jointly. Because of this, our performance will not be bounded by the quality of the VAE decoder. In this paper, we consider VQVAE (Van Den Oord et al., 2017) together with its GAN variant (VQGAN, Esser et al., 2021). For both cases, we transform $256^2 \times 3$ images into $32^2 \times 4$ (i.e., $d_k = 48$) latent space. The VQVAE encoder/decoder is trained on ImageNet (Deng et al., 2009), and is frozen for the rest of the experiments. For $f$-DM-VQGAN, we directly take the checkpoint provided by Rombach et al. (2021). Besides, we need to tune $\gamma_k$ separately for each encoder due to the change in signal magnitude.

# 4 EXPERIMENTS

## 4.1 EXPERIMENTAL SETTINGS

**Datasets.** We evaluate $f$-DMs on five commonly used benchmarks testing generation on a range of domains: FFHQ (Karras et al., 2019), AFHQ (Choi et al., 2020), LSUN Church & Bed (Yu et al., 2015), and ImageNet (Deng et al., 2009). All images are center-cropped and resized to $256 \times 256$.

**Training Details**. We implement the three types of transformations with the same architecture and hyper-parameters except for the stage-specific adapters. We adopt a lighter version of ADM (Dhariwal & Nichol, 2021) as the main U-Net architecture. For all experiments, we adopt the same training scheme using AdamW (Kingma & Ba, 2014) optimizer with a learning rate of $2e-5$ and an EMA decay factor of $0.9999$. We set the weight $\omega_t = \text{sigmoid}(-\log(\alpha_t^2/\sigma_t^2))$ following P2-weighting (Choi et al., 2022). The cosine noise schedule $\alpha_t = \cos(0.5\pi t)$ is adopted for diffusion working in the $256^2 \times 3$ image space. As proposed in Equation 8, noise rescaling (VP by default) is applied for $f$-DMs when the resolutions change. All our models are trained with batch-size 32 images for 500K (FFHQ, AFHQ, LSUN Church), 1.2M (LSUN Bed) and 2.5M (ImageNet) iterations, respectively.

**Baselines & Evaluation.** We compare $f$-DMs against a standard DM (DDPM, Ho et al., 2020) on all five datasets. To ensure a fair comparison, we train DDPM following the same settings and continuous-time formulation as our approaches. We also include transformation-specific baselines: (1) we re-implement the cascaded DM (Cascaded, Ho et al., 2022a) to adapt $f$-DM-DS setup from $16^2$ progressively to $256^2$, where for each stage a separate DM is trained conditioned on the consecutive downsampled image; (2) we re-train a latent-diffusion model (LDM, Rombach et al., 2021) on the extracted latents from our pretrained VQVAE; (3) to compare with $f$-DM-Blur-G, we include the scores and synthesised examples of IHDM (Rissanen et al., 2022). We set 250 timesteps ($\Delta t = 0.004$) for $f$-DMs and the baselines with $\eta = 1$ (Algorithm 1). We use Frechet Inception Distance (FID, Heusel et al., 2017) and Precision/Recall (PR, Kynkäänniemi et al., 2019) as the measures of visual quality, based on 50K samples and the entire training set.

## 4.2 RESULTS

**Qualitative Comparison** To demonstrate the capability of handling various complex datasets, Figure 4 ($\uparrow$) presents an uncurated set of images generated by $f$-DM-DS. We show more samples from all types of $f$-DMs in the Appendix E.4. We also show a comparison between $f$-DMs and the

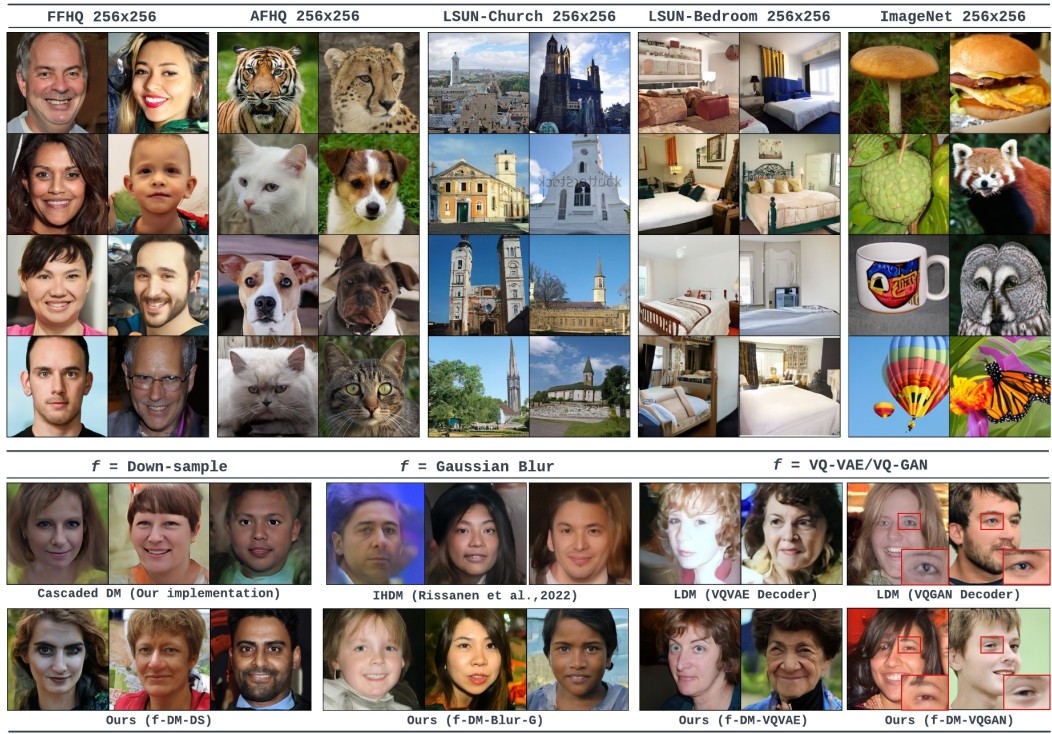

Figure 4: ↑ Random samples from $f$-DM-DS trained on various datasets; ↓ Comparison of $f$-DMs and the corresponding baselines under various transformations. Best viewed when zoomed in. All faces presented are synthesized by the models, and are not real identities.

Table 1: Quantitative comparisons on various datasets. The speed compared to DDPM is calculated with `bsz`= 1 on CPU. Best performing DMs are shown in bold.

| Models | FID↓ | P↑ | R↑ | FID↓ | P↑ | R↑ | Speed | Models | FID↓ |
|---|---|---|---|---|---|---|---|---|---|
| | **FFHQ** 256 × 256 | | | **AFHQ** 256 × 256 | | | | **LSUN-Church** 256 × 256 | |
| DDPM | 10.8 | 0.76 | **0.53** | 9.3 | 0.74 | 0.51 | ×1.0 | DDPM | 9.7 |
| DDPM (1/2) | 16.8 | 0.74 | 0.45 | 15.2 | 0.64 | 0.44 | ×2.0 | $f$-DM-DS | 8.2 |
| | | | | | | | | $f$-DM-VQVAE | **8.0** |
| Cascaded | 49.0 | 0.40 | 0.09 | 24.2 | 0.37 | 0.13 | − | | |
| $f$-DM-DS | 10.8 | 0.74 | 0.50 | 6.4 | **0.81** | 0.48 | ×2.1 | **LSUN-Bed** 256 × 256 | |
| | | | | | | | | DDPM | 8.0 |
| IHDM | 64.9 | − | − | 43.4 | − | − | − | $f$-DM-DS | **6.9** |
| $f$-DM-Blur-G | 11.7 | 0.73 | 0.51 | 6.9 | 0.76 | 0.49 | ×1.0 | $f$-DM-VQVAE | 7.1 |
| $f$-DM-Blur-U | **10.4** | 0.74 | 0.52 | 7.0 | 0.77 | **0.53** | ×1.0 | | |
| LDM | 48.0 | 0.31 | 0.07 | 29.7 | 0.07 | 0.11 | ×9.8 | **ImageNet** 256 × 256 | |
| LDM (GAN)* | 8.6 | 0.72 | 0.60 | 6.5 | 0.63 | 0.61 | ×9.2 | DDPM | 10.9 |
| $f$-DM-VQVAE | 12.7 | **0.77** | 0.47 | 8.9 | 0.76 | 0.40 | ×1.7 | $f$-DM-DS | 8.2 |
| $f$-DM-VQGAN | 11.7 | 0.74 | 0.51 | **5.6** | 0.76 | **0.53** | ×1.7 | $f$-DM-VQVAE | **6.8** |

baselines with various transformations on FFHQ (Figure 4 ↓). Our methods consistently produce better visual results with more coherence and without noticeable artifacts.

**Quantitative Comparison**. We measure the generation quality (FID and precision/recall) and relative inference speed of $f$-DMs and the baselines in Table 1. Across all five datasets, $f$-DMs consistently achieves similar or even better results for the DDPM baselines, while gaining near ×2 inference speed for $f$-DM-{DS, VQVAE, VQGAN} due to the nature of transformations. As a comparison, having fewer timesteps (DDPM 1/2) greatly hurts the generation quality of DDPM. We also show comparisons with transformation-specific baselines on FFHQ & AFHQ.

**v.s. Cascaded DMs**. Although cascaded DMs have been shown effective in literature (Nichol & Dhariwal, 2021; Ho et al., 2022a), it is underexplored to apply cascades in a sequence of consecu-

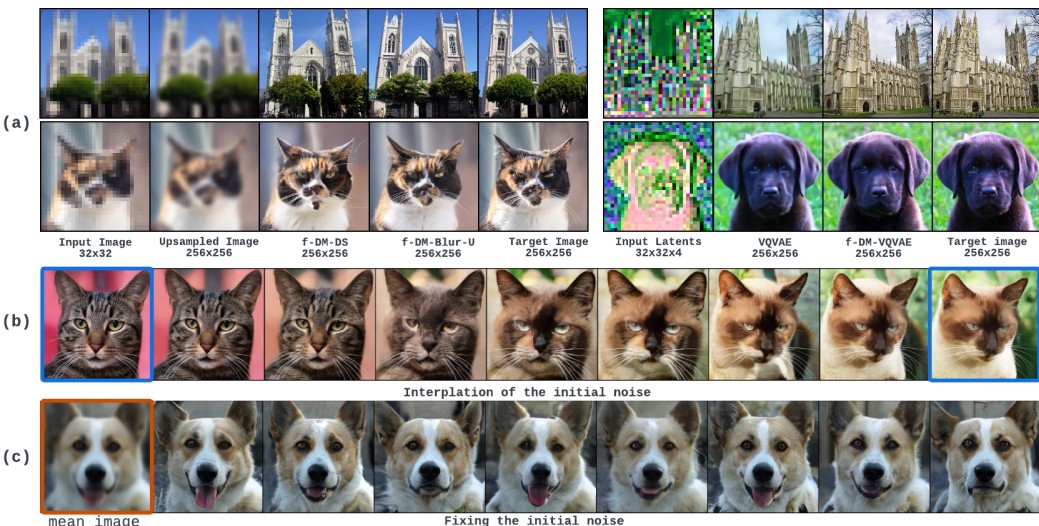

Figure 5: Random DDIM samples ($\eta = 0$) from (a) $f$-DMs on AFHQ and LSUN-Church by given {downsampled, blurred, latent} images as conditions; (b)$f$-DM-VQVAE by interpolating the initial noise of the latent stage; (c)$f$-DM-DS starting from the same initial noise of the $16 \times 16$ stage. For (c), we also show the "mean image" of $300$ random samples using the same initial noise.

tive resolutions ($16 \to 32 \to 64 \to \ldots$) like ours. In such cases, the prediction errors get easily accumulated during the generation, yielding serious artifacts in the final resolution. To ease this, Cascaded DM (Ho et al., 2022a) proposed to apply "noise conditioning augmentation" which reduced the domain gap between stages by adding random noise to the input condition. However, it is not straightforward to tune the noise level for both training and inference time. By contrast, $f$-DM is by-design non-cascaded, and there are no domain gaps between stages. That is, we can train our model end-to-end without worrying the additional tuning parameters and achieve stable results.

**v.s. LDMs**. We show comparisons with LDMs (Rombach et al., 2021) in Table 1. LDMs generate more efficiently as the diffusion only happens in the latent space. However, the generation is heavily biased by the behavior of the fixed decoder. For instance, it is challenging for VQVAE decoders to synthesize sharp images, which causes low scores in Table 1. However, LDM with VQGAN decoders is able to generate sharp details, which are typically favored by InceptionV3 (Szegedy et al., 2016) used in FID and PR. Therefore, despite having artifacts (see Figure 4, below, rightmost) in the output, LDMs (GAN) still obtain good scores. In contrast, $f$-DM, as a pure DM, naturally bridges the latent and image spaces, where the generation is not restricted by the decoder.

**v.s. Blurring DMs**. Table 1 compares with a recently proposed blurring-based method (IHDM, Rissanen et al., 2022). Different from our approach, IHDM formulates a fully deterministic forward process. We conjecture the lack of randomness is the cause of their poor generation quality. Instead, $f$-DM proposes a natural way of incorporating blurring with stochastic noise, yielding better quantitative and qualitative results.

**Conditional Generation**. In Figure 5(a), we demonstrate the example of using pre-trained $f$-DMs to perform conditional generation based on learned transformations. We downsample and blur the sampled real images, and start the reverse diffusion following Section 3.1 with $f$-DM-DS and -Blur-U, respectively. Despite the difference in fine details, both our models faithfully generate high-fidelity outputs close to the real images. The same algorithm is applied to the extracted latent representations. Compared with the original VQVAE output, $f$-DM-VQVAE is able to obtain better reconstruction. We provide additional conditional generation samples with the ablation of the "gradient-based" initialization method in Appendix E.3.

**Latent Space Manipulation**    To demonstrate $f$-DMs have learned certain abstract representations by modeling with signal transformation, we show results of latent manipulation in Figure 5. Here we assume DDIM sampling ($\eta = 0$), and the only stochasticity comes from the initially sampled noise $\epsilon_{\text{full}}$. In (b), we obtain a semantically smooth transition between two cat faces when linearly

interpolating the low-resolution noises; on the other hand, we show samples of the same identity with different fine details (e.g., expression, poses) in (c), which is achieved easily by sampling $f$-DM-DS with the low-resolution ($16^2$) noise fixed. This implies that $f$-DM is able to allocate high-level and fine-grained information in different stages via learning with downsampling.

## 4.3 ABLATION STUDIES

Table 2: Ablation of design choices for $f$-DMs trained on FFHQ. All faces are not real identities.

| Model | Eq. 4 | Rescale | Stages | FID↓ | P↑ | R↑ |
|-------|-------|---------|--------|------|-----|-----|
| $f$-DM-DS | No | VP | cosine | 26.5 | 0.70 | 0.25 |
| | Yes | No | cosine | 14.5 | 0.73 | 0.43 |
| | Yes | SP | cosine | 12.1 | **0.75** | 0.47 |
| | Yes | VP | linear | 13.5 | 0.73 | 0.46 |
| | **Yes** | **VP** | **cosine** | **10.8** | 0.74 | **0.50** |
| $f$-DM-VQVAE | Yes | No | linear | 24.0 | **0.79** | 0.29 |
| | Yes | VP | cosine | 13.8 | 0.78 | 0.45 |
| | **Yes** | **VP** | **linear** | **12.7** | 0.77 | **0.47** |

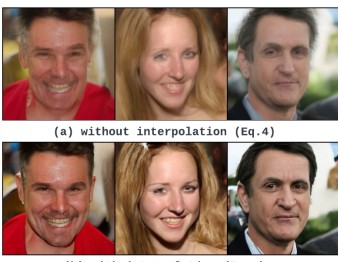

(a) without interpolation (Eq.4)

(b) with interpolation (Eq.4)

Table 2 presents the ablation of the key design choices. As expected, the interpolation formulation (Equation 4) effectively bridges the information gap between stages, without which the prediction errors get accumulated, resulting in blurry outputs and bad scores. Table 2 also demonstrates the importance of applying correct scaling. For both models, rescaling improves the FID and recall by large margins, where SP works slightly worse than VP. In addition, we also empirically explore the difference of stage schedules. Compared to VAE-based models, we usually have more stages in DS/Blur-based models to generate high-resolution images. The *cosine* schedule helps diffusion move faster in regions with low information density (e.g., low-resolution, heavily blurred).

## 5 RELATED WORK

**Progressive Generation with DMs**. Conventional DMs generate images in the same resolutions. Therefore, existing work generally adopt *cascaded* approaches (Nichol & Dhariwal, 2021; Ho et al., 2022a; Saharia et al., 2022a) that chains a series of conditional DMs to generate coarse-to-fine, and have been used in super-resolution (SR3, Saharia et al., 2022b). However, cascaded models tend to suffer error propagation problems. More recently, Ryu & Ye (2022) dropped the need of conditioning, and proposed to generate images in a pyramidal fashion with additional reconstruction guidance; Jing et al. (2022) explored learning subspace DMs and connecting the full space with Langevin dynamics. By contrast, the proposed $f$-DM is distinct from all the above types, which only requires one diffusion process, and the images get naturally up-sampled through reverse diffusion.

**Blurring DMs**. Several concurrent research (Rissanen et al., 2022; Daras et al., 2022; Lee et al., 2022) have recently looked into DM alternatives to combine blurring into diffusion process, some of which also showed the possibility of deterministic generation (Bansal et al., 2022). Although sharing similarities, our work starts from a different view based on signal transformation. Furthermore, our empirical results also show that stochasticity plays a critical role in high-quality generation.

**Latent Space DMs**. Existing work also investigated combining DMs with standard latent variable models. To our best knowledge, most of these works adopt DMs for learning the prior of latent space, where sampling is followed by a pre-trained (Rombach et al., 2021) or jointly optimized (Vahdat et al., 2021) decoder. Conversely, $f$-DM does not rely on the quality decoder.

## 6 CONCLUSION

We proposed $f$-DM, a generalized family of diffusion models that enables generation with signal transformations. As a demonstration, we apply $f$-DM to image generation tasks with a range of transformations, including downsampling, blurring and VAEs, where $f$-DMs outperform the baselines in terms of synthesis quality and semantic interpretation.

ETHICS STATEMENT

Our work focuses on technical development, i,e., synthesizing high-quality images with a range of signal transformations (e.g., downsampling, blurring). Our approach has various applications, such as movie post-production, gaming, helping artists reduce workload, and generating synthetic data as training data for other computer vision tasks. Our approach can be used to synthesize human-related images (e.g., faces), and it is not biased towards any specific gender, race, region, or social class.

However, the ability of generative models, including our approach, to generate high-quality images that are indistinguishable from real images, raises concerns about the misuse of these methods, e.g., generating fake images. To resolve these concerns, we need to mark all the generated results as "synthetic". In addition, we believe it is crucial to have authenticity assessment, such as fake image detection and identity verification, which will alleviate the potential for misuse. We hope our approach can be used to foster the development of technologies for authenticity assessment. Finally, we believe that creating a set of appropriate regulations and laws would significantly reduce the risks of misuse while bolstering positive effects on technology development.

REPRODUCIBILITY STATEMENT

We assure that all the results shown in the paper and supplemental materials can be reproduced. We believe we have provided enough implementation details in the paper and supplemental materials for the readers to reproduce the results.

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

# APPENDIX

# A  DETAILED DERIVATION OF $f$-DMs

## A.1  $q(\mathbf{z}_t|\mathbf{z}_s, \mathbf{x})$

We derive the definition in Equation 5 with the change-of-variable trick given the fact that $\boldsymbol{x}_t, \boldsymbol{x}_s$ and $\boldsymbol{x}^k$ are all deterministic functions of $\boldsymbol{x}$.

More precisely, suppose $\boldsymbol{z}_t \sim \mathcal{N}(\alpha_t \boldsymbol{x}_t, \sigma_t^2 I), \boldsymbol{z}_s \sim \mathcal{N}(\alpha_s \boldsymbol{x}_s, \sigma_s^2 I)$, where $\tau_k \leq s < t < \tau_{k+1}$. Thus, it is equivalent to have $\boldsymbol{u}_t \sim \mathcal{N}(\alpha_t \boldsymbol{x}^k, \sigma_t^2 I), \boldsymbol{u}_s \sim \mathcal{N}(\alpha_s \boldsymbol{x}^k, \sigma_s^2 I), \boldsymbol{u}_t = \boldsymbol{z}_t - \alpha_t(\boldsymbol{x}_t - \boldsymbol{x}^k), \boldsymbol{u}_s = \boldsymbol{z}_s - \alpha_s(\boldsymbol{x}_s - \boldsymbol{x}^k)$. From the above definition, it is reasonable to assume $\boldsymbol{u}_t, \boldsymbol{u}_s$ follow the standard DM transitionm which means that:

$$\boldsymbol{u}_t = \alpha_{t|s} \boldsymbol{u}_s + \sigma_{t|s}\boldsymbol{\epsilon}, \ \ \boldsymbol{\epsilon} \sim \mathcal{N}(\mathbf{0}, I)$$
$$\Rightarrow \boldsymbol{z}_t - \alpha_t(\boldsymbol{x}_t - \boldsymbol{x}^k) = \alpha_{t|s}\left(\boldsymbol{z}_s - \alpha_s(\boldsymbol{x}_s - \boldsymbol{x}^k)\right) + \sigma_{t|s}\boldsymbol{\epsilon}, \ \ \boldsymbol{\epsilon} \sim \mathcal{N}(\mathbf{0}, I)$$
$$\Rightarrow \qquad \boldsymbol{z}_t = \alpha_{t|s}\boldsymbol{z}_s + \alpha_t(\boldsymbol{x}_t - \boldsymbol{x}_s) + \sigma_{t|s}\boldsymbol{\epsilon}, \ \ \boldsymbol{\epsilon} \sim \mathcal{N}(\mathbf{0}, I)$$

As typically $\boldsymbol{x}_t \neq \boldsymbol{x}_s$ and both $\boldsymbol{x}_t, \boldsymbol{x}_s$ are the functions of $\boldsymbol{x}^k$. Then $\boldsymbol{z}_t$ is dependent on both $\boldsymbol{z}_s$ and $\boldsymbol{x}^k = f_{0:k}(\boldsymbol{x})$, resulting in a non-Markovian transition:

$$q(\boldsymbol{z}_t|\boldsymbol{z}_s, \boldsymbol{x}) = \mathcal{N}(\boldsymbol{z}_t; \alpha_{t|s}\boldsymbol{z}_s + \alpha_t \cdot (\boldsymbol{x}_t - \boldsymbol{x}_s), \sigma_{t|s}^2 I),$$

Note that, this equation stands only when $\boldsymbol{x}_t, \boldsymbol{x}_s$ and $\boldsymbol{x}_k$ in the same space, and we did not make specific assumptions to the form of $\boldsymbol{x}_t$.

## A.2  $q(\mathbf{z}_s|\mathbf{z}_t, \mathbf{x})$

The reverse diffusion distribution follows the Bayes' Theorem: $q(\boldsymbol{z}_s|\boldsymbol{z}_t, \boldsymbol{x}) \propto q(\boldsymbol{z}_s|\boldsymbol{x})q(\boldsymbol{z}_t|\boldsymbol{z}_s, \boldsymbol{x})$, where both $q(\boldsymbol{z}_s|\boldsymbol{x})$ and $q(\boldsymbol{z}_t|\boldsymbol{z}_s, \boldsymbol{x})$ are Gaussian distributions with general forms of $\mathcal{N}(\boldsymbol{z}_s|\boldsymbol{\mu}, \sigma^2 I)$ and $\mathcal{N}(\boldsymbol{z}_t|A\boldsymbol{z}_s + \boldsymbol{b}, \sigma'^2 I)$, respectively. Based on Bishop & Nasrabadi (2006) (2.116), we can derive:

$$q(\boldsymbol{z}_s|\boldsymbol{z}_t, \boldsymbol{x}) = \mathcal{N}(\boldsymbol{z}_s|\bar{\sigma}^{-2}\left(\sigma'^{-2}A^\top(\boldsymbol{z}_t - \boldsymbol{b}) + \sigma^{-2}\boldsymbol{\mu}\right), \bar{\sigma}^2 I),$$

where $\bar{\sigma}^2 = (\sigma^{-2} + \sigma'^{-2}\|A\|^2)^{-1}$. Therefore, we can get the exact form by plugging our variables $\boldsymbol{\mu} = \alpha_s \hat{\boldsymbol{x}}_k^s, \sigma = \sigma_s, A = \alpha_{t|s}I, \boldsymbol{b} = \alpha_t \cdot (\boldsymbol{x}_t - \boldsymbol{x}_s), \sigma' = \sigma_{t|s}$ into above equation, we get:

$$q(\boldsymbol{z}_s|\boldsymbol{z}_t, \boldsymbol{x}) = \mathcal{N}(\boldsymbol{z}_s|\alpha_s \boldsymbol{x}_s + \sqrt{\sigma_s^2 - \bar{\sigma}^2}\boldsymbol{\epsilon}_t, \bar{\sigma}^2 I),$$

where $\boldsymbol{\epsilon}_t = (\boldsymbol{z}_t - \alpha_t \boldsymbol{x}_t)/\sigma_t$ and $\bar{\sigma} = \sigma_s \sigma_{t|s}/\sigma_t$.

Alternatively, if we assume $\boldsymbol{x}_t$ take the interpolation formulation in Equation 4, we can also re-write $\boldsymbol{x}_s$ with $\boldsymbol{x}_t + \frac{t-s}{t-\tau_k}\boldsymbol{\delta}_t$, where we define a new variable $\boldsymbol{\delta}_t = \boldsymbol{x}^k - \boldsymbol{x}_t$. As stated in the main context (Section 3.1), such change makes $q(\boldsymbol{z}_t|\boldsymbol{z}_s, \boldsymbol{x})$ avoid computing $\boldsymbol{x}_s$ which may be potentially costly. In this way, we re-write the above equation as follows:

$$q(\boldsymbol{z}_s|\boldsymbol{z}_t, \boldsymbol{x}) = \mathcal{N}(\boldsymbol{z}_s|, \alpha_s(\boldsymbol{x}_t + \boldsymbol{\delta}_t \cdot (t - s)/(t - \tau_k)) + \sqrt{\sigma_s^2 - \bar{\sigma}^2}\boldsymbol{\epsilon}_t, \bar{\sigma}^2 I), \qquad (10)$$

## A.3  DIFFUSION INSIDE STAGES

In the inference time, we generate data by iteratively sampling from the conditional distribution $p(\boldsymbol{z}_s|\boldsymbol{z}_t) = \mathbb{E}_{\boldsymbol{x}}[q(\boldsymbol{z}_s|\boldsymbol{z}_t, \boldsymbol{x})]$ based on Equation 10. In practice, the expectation over $\boldsymbol{x}$ is approximated by our model's prediction. As shown in Equation 9, in this work, we propose a "double-prediction" network $\theta$ that reads $\boldsymbol{z}_t$, and simultaneously predicts $\boldsymbol{x}_t$ and $\boldsymbol{\delta}_t$ with $\boldsymbol{x}_\theta$ and $\boldsymbol{\delta}_\theta$, respectively. The predicted Gaussian noise is denoted as $\boldsymbol{\epsilon}_\theta = (\boldsymbol{z}_t - \alpha_t \boldsymbol{x}_\theta)/\sigma_t$. Note that the prediction $\boldsymbol{x}_\theta$ and $\boldsymbol{\epsilon}_\theta$ are interchangable, which means that we can readily derive one from the other's prediction. Therefore, by replacing $\boldsymbol{x}_t, \boldsymbol{\delta}_t, \boldsymbol{\epsilon}_t$, with $\boldsymbol{x}_\theta, \boldsymbol{\delta}_\theta, \boldsymbol{\epsilon}_\theta$ in Equation 10, we obtain the sampling algorithm shown in Algorithm 1: Line 6.

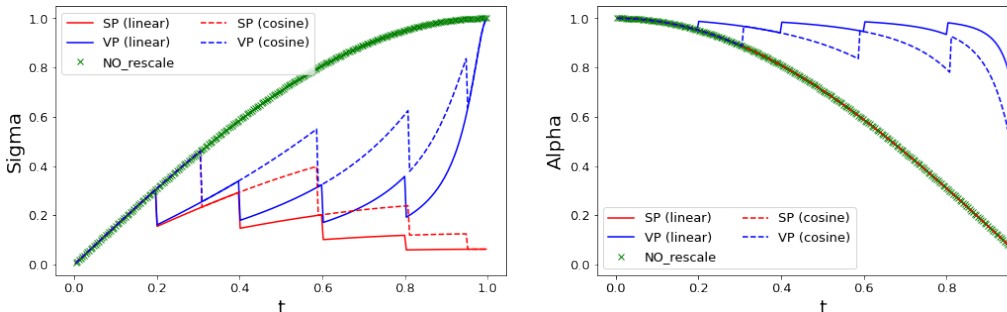

Figure 7: Illustration of noise schedule ($\alpha_t$ and $\sigma_t$) for $f$-DM-DS models with 5 stages ($16^2 \rightarrow 256^2$). We use the standard cosine noise schedule $\alpha_t = \cos(0.5\pi t)$. We also show the difference between the linear/cosine stage schedule, as well as the proposed SP/VP re-scaling methods.

## A.4 NOISE AT BOUNDARIES

In this paper, the overall principle is to handle the transition across stage boundary is to ensure the forward diffusion to be deterministic and smooth, therefore almost no information is lost during the stage change. Such requirement is important as it directly correlated to the denoising performance. Failing to recover the lost information will directly affect the diversity of the model generates.

**Forward diffusion** As described in Section 3.1, since we have the control of the signal and the noise separately, we can directly apply the deterministic transformation on the signal, and dropping noise elements.

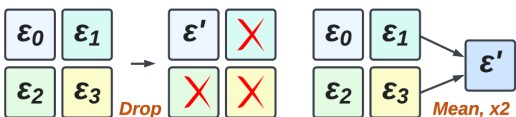

Figure 6: Two naïve ways for down-sampling.

Alternatively, we also implemented a different $\zeta(\epsilon)$ based on averaging. As shown in Figure 6, if the transformation is down-sampling, we can use the fact that the mean of Gaussian noises is still Gaussian with lower variance: $(\epsilon_0 + \epsilon_1 + \epsilon_2 + \epsilon_3)/4 \sim \mathcal{N}(\mathbf{0}, \frac{1}{4}I)$. Therefore, $\times 2$ rescaling is needed on the resulted noise.

**Reverse diffusion** Similarly, we can also define the reverse process if $\zeta$ is chosen to be averaging. Different from "dropping" where the reverse process is simply adding independent Gaussian noises, the reverse of "averaging" requests to sample $\sum_{i=0}^{3} \epsilon_i = 2\epsilon$ given the input noise $\epsilon$, while having $p(\epsilon_i) = \mathcal{N}(\mathbf{0}, I), i = 0, 1, 2, 3$. Such problem has a closed solution and can be implemented in an autoregressive fashion:

$$a = 2\epsilon;$$
$$\epsilon_0 = a/4 + \sqrt{3/4} \cdot \hat{\epsilon}_1, \ a = a - \epsilon_0, \ \hat{\epsilon}_1 \sim \mathcal{N}(\mathbf{0}, I);$$
$$\epsilon_1 = a/3 + \sqrt{2/3} \cdot \hat{\epsilon}_2, \ a = a - \epsilon_1, \ \hat{\epsilon}_2 \sim \mathcal{N}(\mathbf{0}, I);$$
$$\epsilon_2 = a/2 + \sqrt{1/2} \cdot \hat{\epsilon}_3, \ a = a - \epsilon_2, \ \hat{\epsilon}_3 \sim \mathcal{N}(\mathbf{0}, I);$$
$$\epsilon_3 = a$$

Similar to the case of "dropping", we also need 3 additional samples $\hat{\epsilon}_{1:3}$ to contribute to four noises, therefore it can be implemented in the same way as described in Section 3.1. Empirically, reversing the "averaging" steps tends to produce samples with better FID scores. However, since it introduces correlations into the added noise, which may cause undesired biases especially in DDIM sampling.

**Intuition behind Re-scaling** Here we present a simple justification of applying noise rescaling. Suppose the signal dimensionality changes from $M_{k-1}$ to $M_k$ when crossing the stage boundary, and such change is caused by different sampling rates. Based the proposed resolution-agnostic SNR (Equation 7), the number of sampled points inside $\Omega$ is proportional to its dimensionality. Generally, it is safe to assume signals are mostly low-frequency. Therefore, averaging signals will not change its variance. By contrast, as shown above, averaging Gaussian noises results in lower variance, where

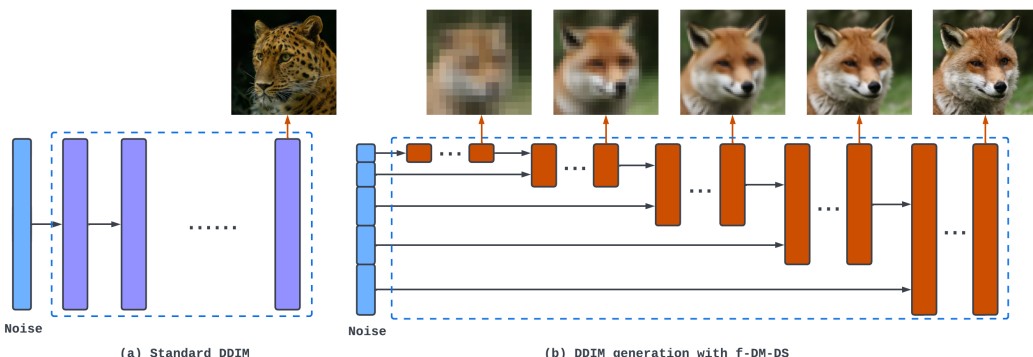

Figure 8: We show the comparison of the DDIM sampling.

in our case, the variance is proportional to $M^{-1}$. Therefore, suppose the signal magnitude does not change, we can get the re-scaling low by forcing $\mathrm{SNR}(\boldsymbol{z}_\tau) = \mathrm{SNR}(\boldsymbol{z}_{\tau^-})$ at the stage boundary:

$$\sigma_{\tau^-}^2 \cdot M_{k-1}^{-1} = \sigma_\tau^2 \cdot M_k^{-1},$$

which derives the signal preserving (SP) rescaling in Equation 8. In Figure 7, we show an example of the change of $\alpha$ and $\sigma$ with and without applying the re-scaling techqnique for $f$-DM-DS models.

## A.5 DDIM SAMPLING

The above derivations only describe the standard ancestral sampling ($\eta = 1$) where $q(\boldsymbol{z}_s | \boldsymbol{z}_t, \boldsymbol{x})$ is determined by Bayes' Theorem. Optionally, one can arbitrarily define any proper reverse diffusion distribution as long as the marginal distributions match the definition. For example, $f$-DM can also perform deterministic DDIM (Song et al., 2021a) by setting $\eta = 0$ in Algorithm 1. Similar to Song et al. (2021a), we can also obtain the proof based on the induction argument.

Figure 8 shows the comparison of DDIM sampling between the standard DMs and the proposed $f$-DM. In DDIM sampling ($\eta = 0$), the only randomness comes from the initial noise at $t = 1$. Due to the proposed noise resampling technique, $f$-DM enables a multi-scale noising process where the sampled noises are splitted and sent to different steps of the diffusion process. In this case, compared to standard DMs, we gain the ability of controlling image generation at different levels, resulting in smooth semantic interpretation.

## B DETAILED INFORMATION OF TRANSFORMATIONS

We show the difference of all the transformations used in this paper in Figure 9.

### B.1 DOWNSAMPLING

In early development of this work, we explored various combinations of performing down-sampling: $\boldsymbol{f} = \{\text{bilinear, nearest, Gaussian blur + subsample}\}$, $\boldsymbol{g} = \{\text{bilinear, bicubic, nearest, neural-based}\}$. While all these combinations produced similar results, we empirically on FFHQ found that both choosing *bilinear interpolation* for both $\boldsymbol{f}, \boldsymbol{g}$ achieves most stable results. Therefore, all the main experiments of $f$-DM-DS are conducted on bilinear interpolation. As discussed in Section 3.2, we choose $K = 4$, which progressively downsample a $256^2$ into $16^2$.

### B.2 BLURRING

We experimented two types of blurring functions. For upsampling-based blurring, we use the same number of stages as the downsampling case; for Gaussian-based blurring, we adopt $K = 7$ with corresponding kernel sizes $\sigma_B = 15 \sin^2(\frac{\pi}{2}\tau_k)$, where $\tau_k$ follows the *cosine* stage schedule. In practice, we implement blurring function in frequency domain following Rissanen et al. (2022) based on discrete cosine transform (DCT).

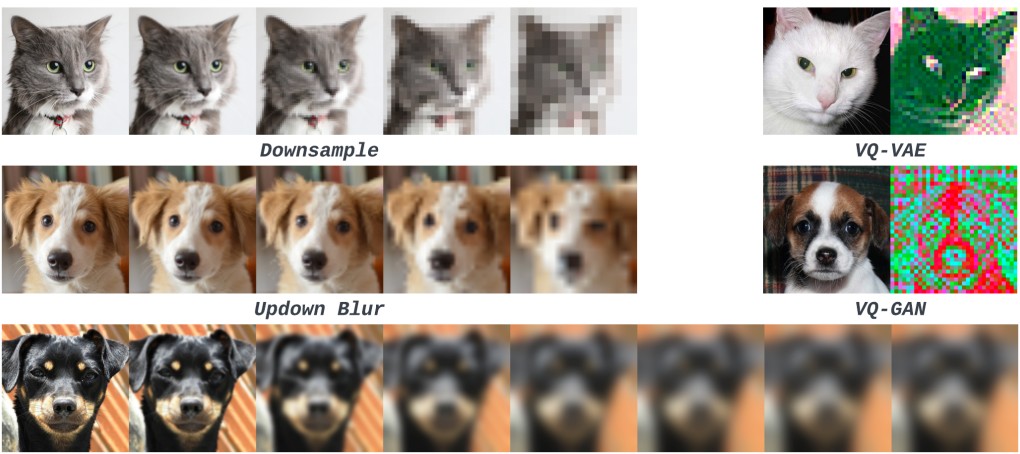

Figure 9: We show examples of the five transformations (downsample, blur, VAEs) used in this paper. For downsampling, we resize the image with nearest upsampler; for VQ-VAE/VQ-GAN, we visualize the first 3 channels of the latent feature maps.

### B.3 VAEs

In this paper, we only consider vector quantized (VQ) models with single layer latent space, while our methods can be readily applied to hierarchical (Razavi et al., 2019) and KL-regularized VAE models (Vahdat & Kautz, 2020). Following Rombach et al. (2021), we take the feature vectors before the quantization layers as the latent space, and keep the quantization step in the decoder ($g$) when training diffusion models.

We follow an open-sourced implementation [2] to train our VQVAE model on ImageNet. The model consists of two strided convolution blocks which by default downsamples the input image by a factor of $8$. We use the default hyper-parameters and train the model for $50$ epochs with a batch-size of $128$. For a fair comparison to match the latent size of VQVAE, we use the pre-trained autoencoding model (Rombach et al., 2021) with the setting of {f=8, VQ (Z=256, d=4)}. We directly use the checkpoint [3] provided by the authors. Note that the above setting is not the best performing model (LDM-4) in the original paper. Therefore, it generates more artifacts when reconstructing images from the latents.

Before training, we compute the signal magnitude ratio $\gamma_k$ (Equation 8) over the entire training set of FFHQ, where we empirically set $\gamma_k = 2.77$ for VQ-GAN and $\gamma_k = 2.0$ for VQ-VAE, respectively.

## C  DATASET DETAILS

**FFHQ** (https://github.com/NVlabs/ffhq-dataset) contains 70k images of real human faces in resolution of $1024^2$. For most of our experiments, we resize the images to $256^2$.

**AFHQ** (https://github.com/clovaai/stargan-v2#animal-faces-hq-dataset-afhq) contains 15k images of animal faces including cat, dog and wild three categories in resolution of $512^2$. We train conditional diffusion models by merging all training images with the label information. All images are resized to $256^2$.

**LSUN** (https://www.yf.io/p/lsun) is a collection of large-scale image dataset containing 10 scenes and 20 object categories. Following previous works Rombach et al. (2021), we choose the two categories – Church (126k images) and Bed (3M images), and train separate unconditional models on them. As LSUN-Bed is relatively larger, we set the iterations longer than other datasets. All images are resized to $256^2$ with center-crop.

---

[2] https://github.com/rosinality/vq-vae-2-pytorch
[3] https://ommer-lab.com/files/latent-diffusion/vq-f8-n256.zip

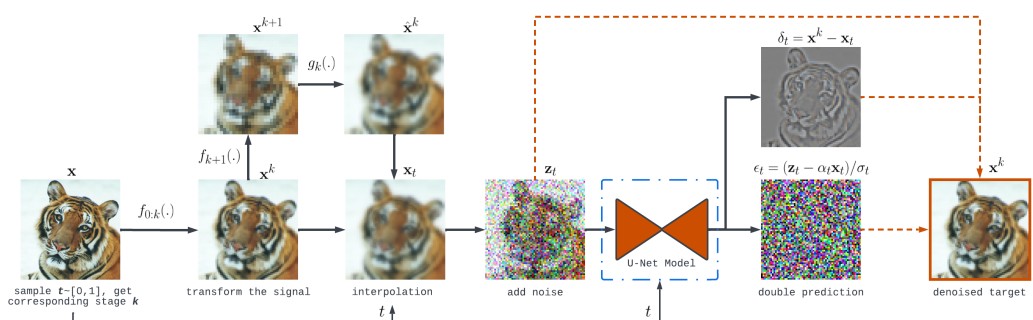

Figure 10: An illustration of the training pipeline.

**ImageNet** (`https://image-net.org/download.php`) we use the standard ImageNet-1K dataset which contains 1.28M images across 1000 classes. We directly merge all the training images with class-labels. All images are resized to $256^2$ with center-crop. For both $f$-DM and the baseline models, we adopt the classifier-free guidance (Ho & Salimans, 2022) with the unconditional probability $0.2$. In the inference time, we use the guidance scale ($s = 2$) for computing FIDs, and $s = 3$ to synthesize examples for comparison.

## D  IMPLEMENTATION DETAILS

### D.1  ARCHITECTURE CONFIGURATIONS

We implement $f$-DM strictly following standard U-Net architecture in Nichol & Dhariwal (2021). As shown in Figure 11, input $z_t$ will be directed to the corresponding inner layer based on spatial resolutions, and a stage-specific adapter is adopted to transform the channel dimension. Such architecture also allows memory-efficient batching across stages where we can create a batch with various resolutions, and split the computation based on the resolutions.

### D.2  HYPER-PARAMETERS

In our experiments, we adopt the following two sets of parameters based on the complexity of the dataset: *base* (FFHQ, AFHQ, LSUN-Church/Bed) and *big* (ImageNet). For *base*, we use 1 residual block per resolution, with the basic dimension 128. For *big*, we use 2 residual blocks with the basic dimension 192. Given one dataset, all the models with various transformations including the baseline DMs share the same hyper-parameters except for the adapters. We list the hyperparameter details in Table 3.

| Hyper-param. | FFHQ | AFHQ | LSUN-Church | LSUN-Bed | ImageNet |
|---|---|---|---|---|---|
| image res. | $256^2$ | $256^2$ | $256^2$ | $256^2$ | $256^2$ |
| # of classes | None | 3 | None | None | 1000 |
| c.f. guidance | - | No | - | - | Yes |
| #channels | 128 | 128 | 128 | 128 | 192 |
| #res-blocks | 1 | 1 | 1 | 1 | 2 |
| channel multi. | | | $[1, 1, 2, 2, 4, 4]$ | | |
| attention res. | | | $16, 8$ | | |
| batch size | 32 | 32 | 32 | 32 | 64 |
| lr | | | $2e{-}5$ | | |
| iterations | $500K$ | $500K$ | $500K$ | $1200K$ | $2500K$ |

Table 3: Hyperparameters and settings for $f$-DM on different datasets.

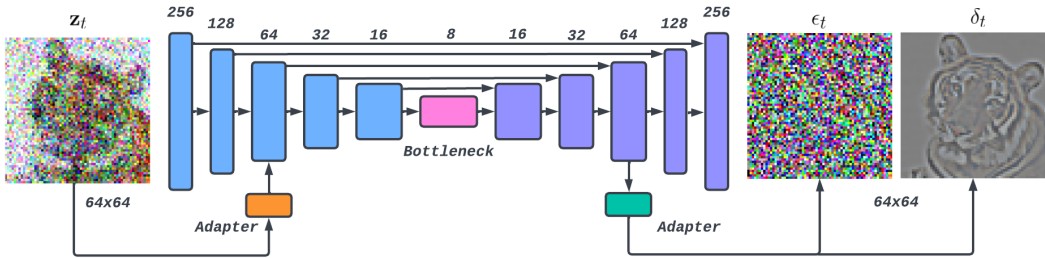

Figure 11: An illustration of the modified U-Net architecture. Time conditioning is omitted. The parameters are partially shared across stages based on the resolutions. Stage-specific adapters are adopted to transform the input dimensions.

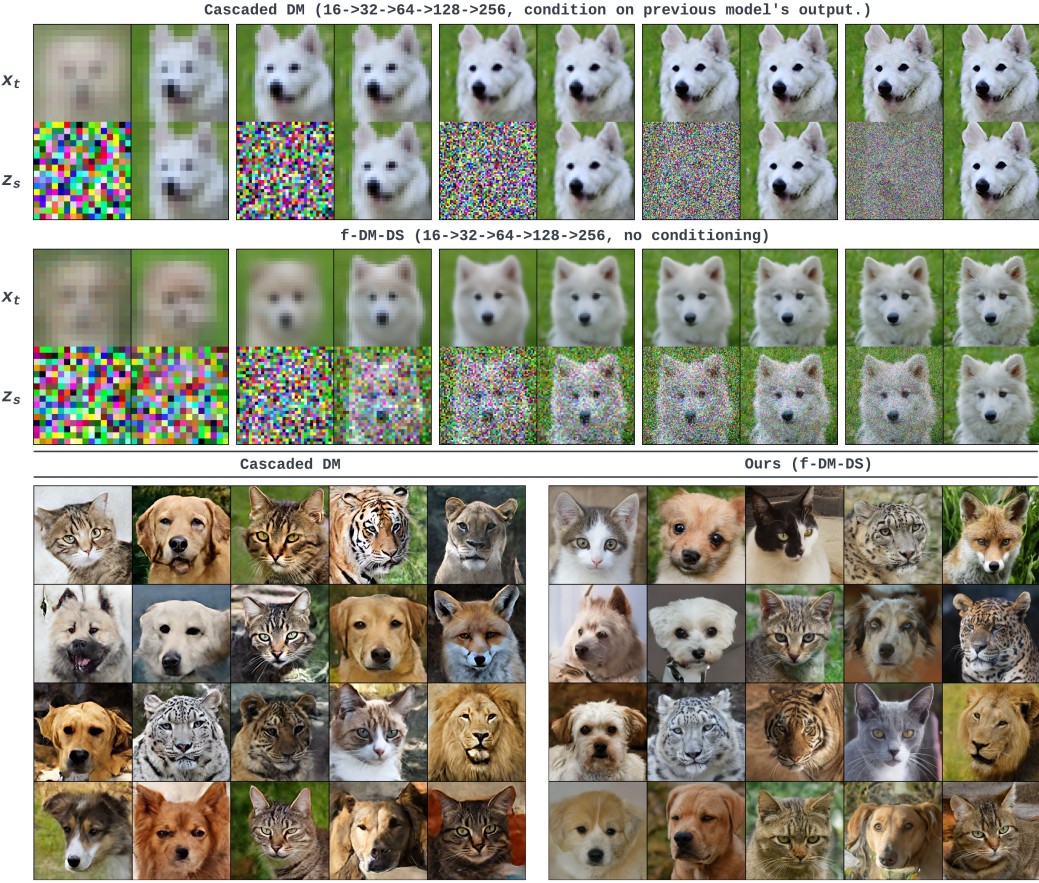

Figure 12: Additional comparisons with Cascaded DM on AFHQ. ↑ Comparison of the reverse diffusion process from $16^2$ to $256^2$. We visualize the denoised outputs ($x_t$) and the corresponding next noised input ($z_s$) near the start & end of each resolution diffusion. ↓ Comparison of random samples generated by Cascaded DM and $f$-DM-DS.

# E  ADDITIONAL RESULTS

## E.1  QUANTITATIVE COMPARISON WITH DDIM

We also include comparison of $f$-DM with the standard DM using DDIM sampling ($\eta = 0$) in Table 4. Similar to the conclusion drawn from Table 1, the proposed $f$-DM can achieve comparable or even better performance than baseline DM even with $\eta = 0$ (generation only controlled by the initial noise, see Figure 8), while having better scores for DDIM with half generation steps.

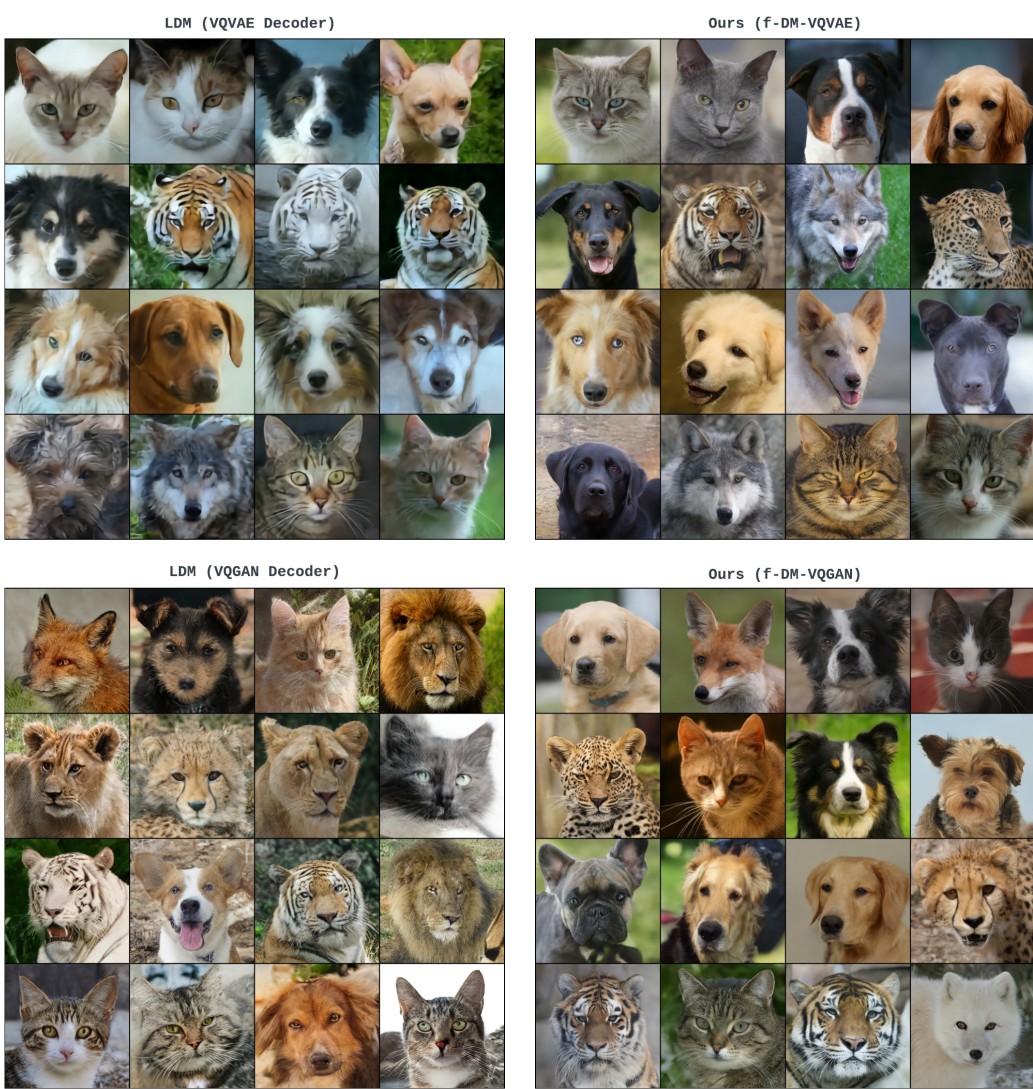

Figure 13: Additional comparisons with LDMs on AFHQ.

| Models | FID↓ | P↑ | R↑ | FID↓ | P↑ | R↑ | Speed |
|---|---|---|---|---|---|---|---|
| | **FFHQ** $256 \times 256$ | | | **AFHQ** $256 \times 256$ | | | |
| DDIM | **11.4** | 0.71 | 0.53 | 12.1 | 0.58 | **0.65** | ×1.0 |
| DDIM (1/2) | 13.0 | 0.70 | 0.51 | 16.8 | 0.48 | 0.64 | ×2.0 |
| $f$-DM-DS ($\eta = 0$) | 12.6 | **0.76** | **0.55** | **5.8** | **0.76** | 0.55 | ×2.1 |

Table 4: Comparison on FFHQ and AFHQ for DDIM sampling ($\eta = 0$)

## E.2 V.S. TRANSFORMATION-SPECIFIC BASELINES

We include more comparisons in Figure 12 and 13. From Figure 12, we compare the generation process of $f$-DM and the cascaded DM. It is clear that $f$-DM conducts coarse-to-fine generation in a more natural way, and the results will not suffer from error propagation. As shown in Figure 13, LDM outputs are easily affected by the chosen decoder. VQVAE decoder tends output blurry images; the output from VQGAN decoder has much finer details while remaining noticable artifacts (e.g., eyes, furs). By contrast, $f$-DM perform stably for both latent spaces.

### E.3 CONDITIONAL GENERATION

We include additional results of conditional generation, i.e., super-resolution (Figure 14) and de-blurring (Figure 15). We also show the comparison with or without the proposed gradient-based initialization, which greatly improves the faithfulness of conditional generation when the input noise is high (e.g., $16 \times 16$ input).

### E.4 ADDITIONAL QUALITATIVE RESULTS

Finally, we provide additional qualitative results for our unconditional models for FFHQ (Figure 16), AFHQ (Figure 17), LSUN (Figure 18) and our class-conditional ImageNet model (Figure 19,20).

## F LIMITATIONS AND FUTURE WORK

Although $f$-DM enables diffusion with signal transformations, which greatly extends the scope of DMs to work in transformed space, there still exist limitations and opportunities for future work. First, it is an empirical question to find the optimal stage schedule for all transformations. Our ablation studies also show that different heuristics have differences for DS-based and VAE-based models. A metric that can automatically determine the best stage schedule based on the property of each transformation is needed and will be explored in the future. In addition, although the current method achieves faster inference when generating with transformations like down-sampling, the speed-up is not very significant as we still take the standard DDPM steps. How to further accelerate the inference process of DMs is a challenging and orthogonal direction. For example, it has great potential to combine $f$-DM with speed-up techniques such as knowledge distillation (Salimans & Ho, 2022). Moreover, no matter hand-designed or learned, all the transformations used in $f$-DM are still fixed when training DM. It is, however, different from typical VAEs, where both the encoder and decoder are jointly optimized during training. Therefore, starting from a random/imperfect transformation and training $f$-DM jointly with the transformations towards certain target objectives will be studied as future work.

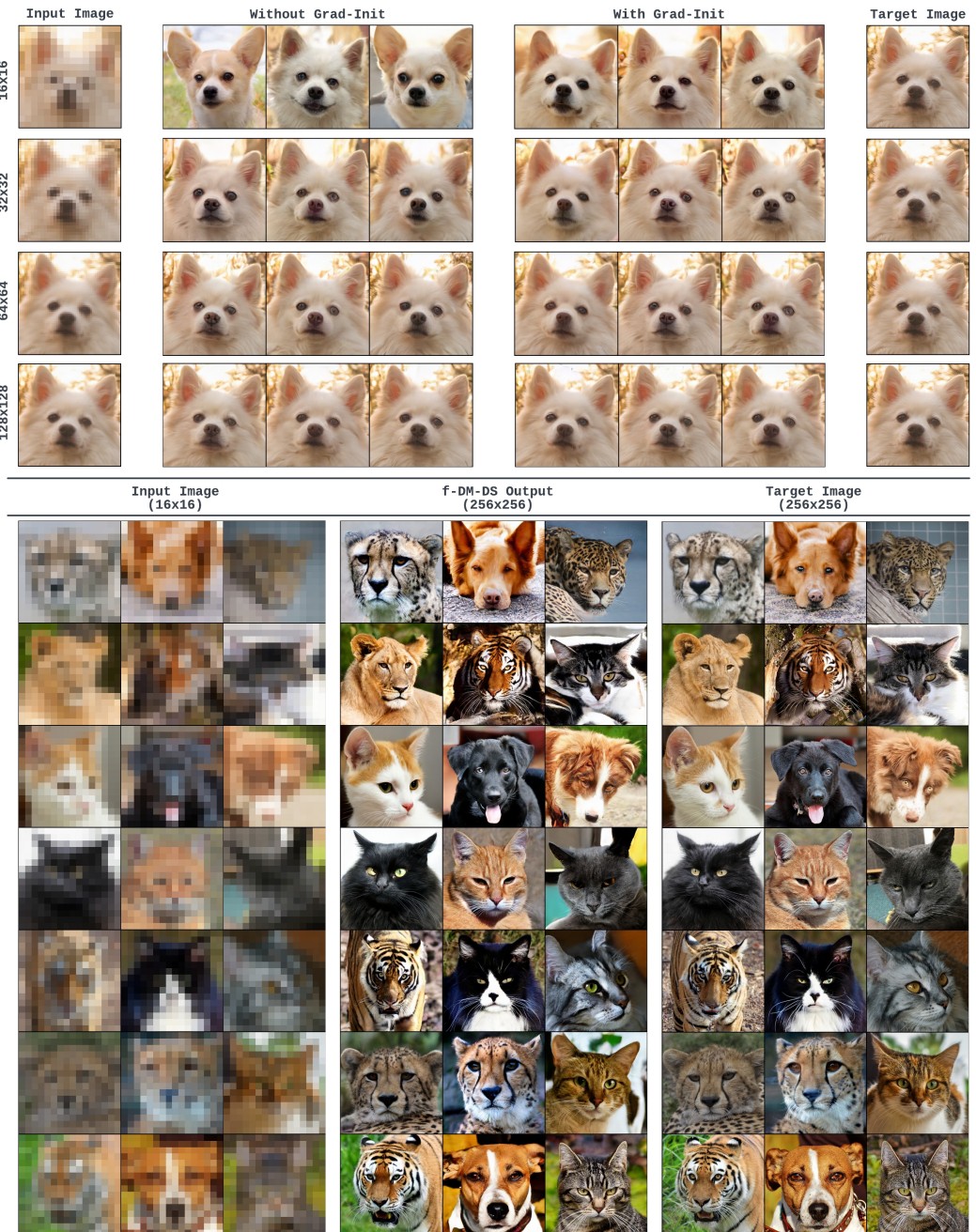

Figure 14: Additional examples of super-resolution (SR) with the unconditional $f$-DM-DS trained on AFHQ. ↑ The same input image with various resolution $16^2, 32^2, 64^2, 128^2$. We sample 3 random seeds for each resolution input. We also show the difference with and without applying gradient-based initialization (Grad-Init) on $z$. ↓ SR results of various $16^2$ inputs.

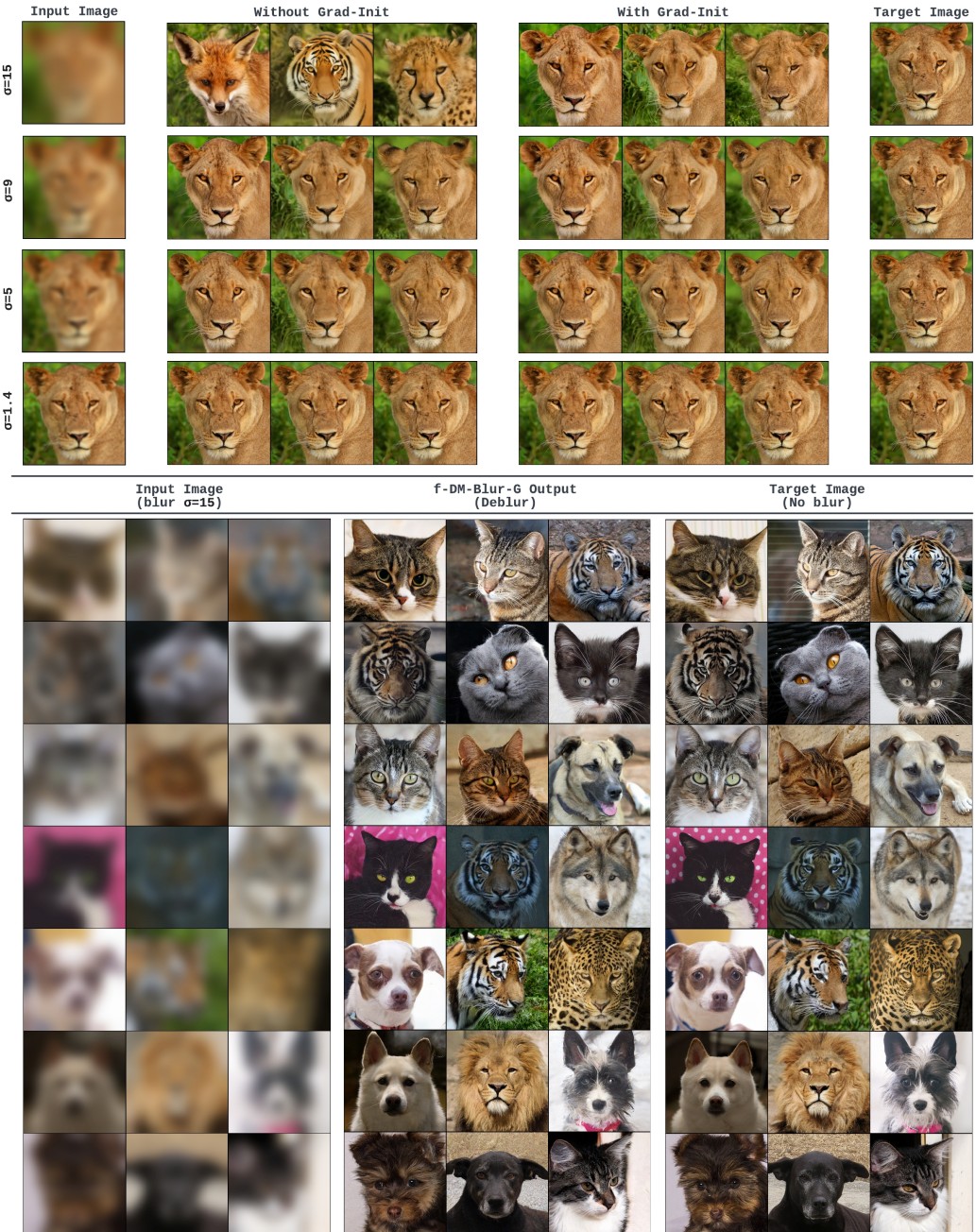

Figure 15: Additional examples of de-blurring with the unconditional $f$-DM-Blur-G trained on AFHQ. ↑ The same input image with various Gaussian kernel sizes $\sigma = 15, 9, 4, 1.4$. We sample 3 random seeds for each resolution input. We also show the difference with and without applying gradient-based initialization (Grad-Init) on $z$. ↓ Deblurred results of various blur images.

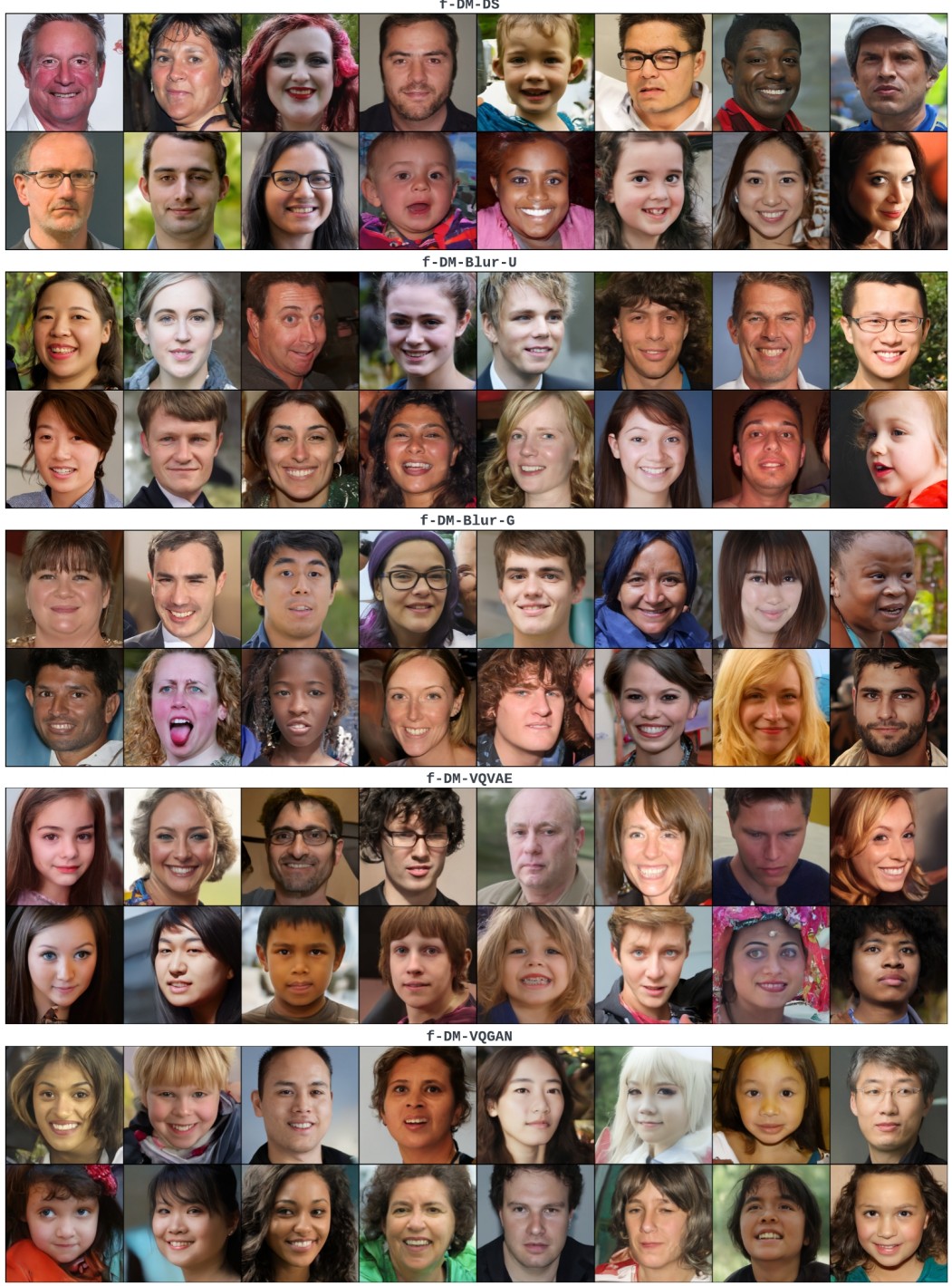

Figure 16: Random samples generated by five $f$-DMs trained on FFHQ $256 \times 256$. All faces presented are synthesized by the models, and are not real identities.

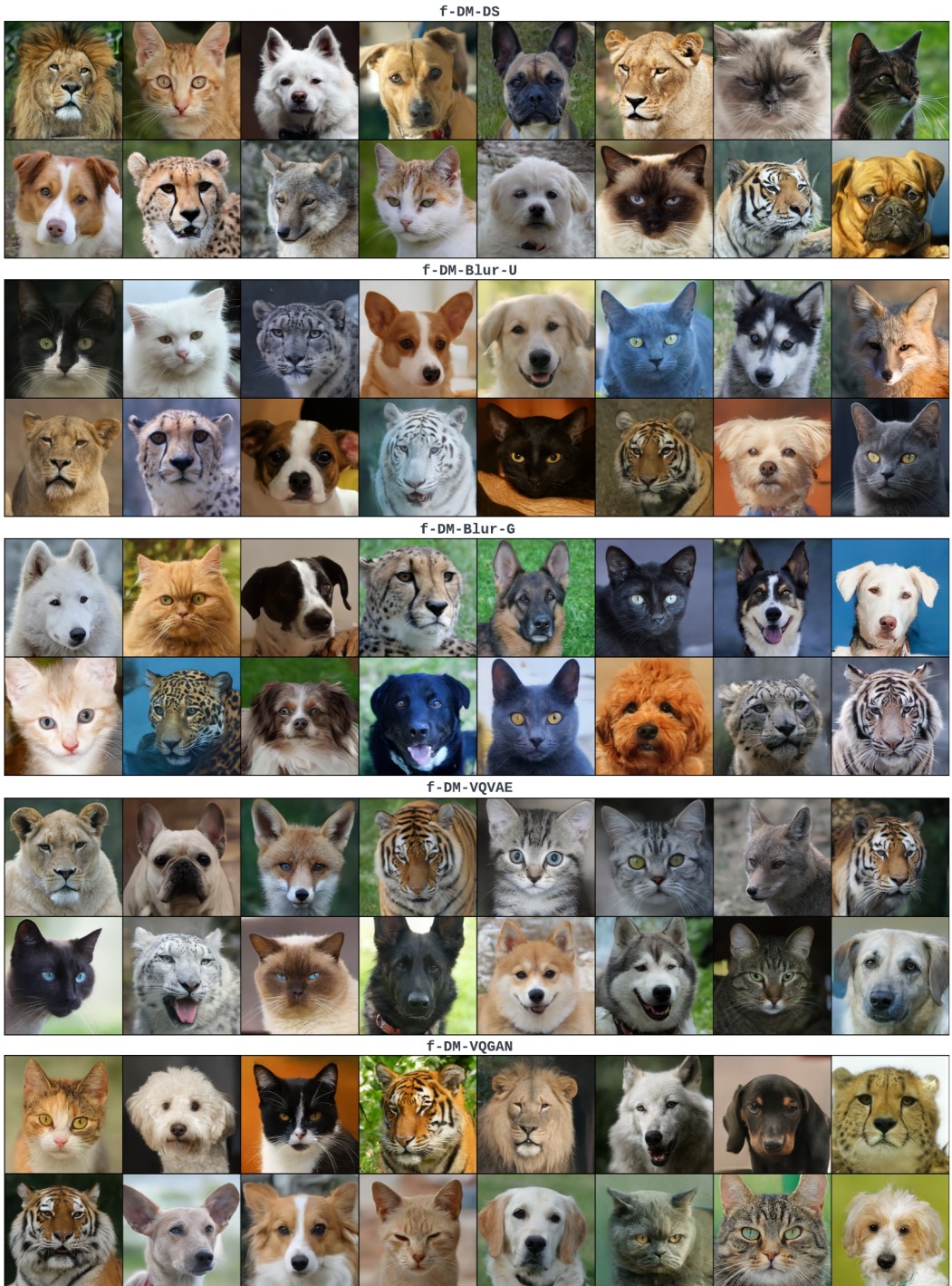

Figure 17: Random samples generated by five $f$-DMs trained on AFHQ $256 \times 256$.

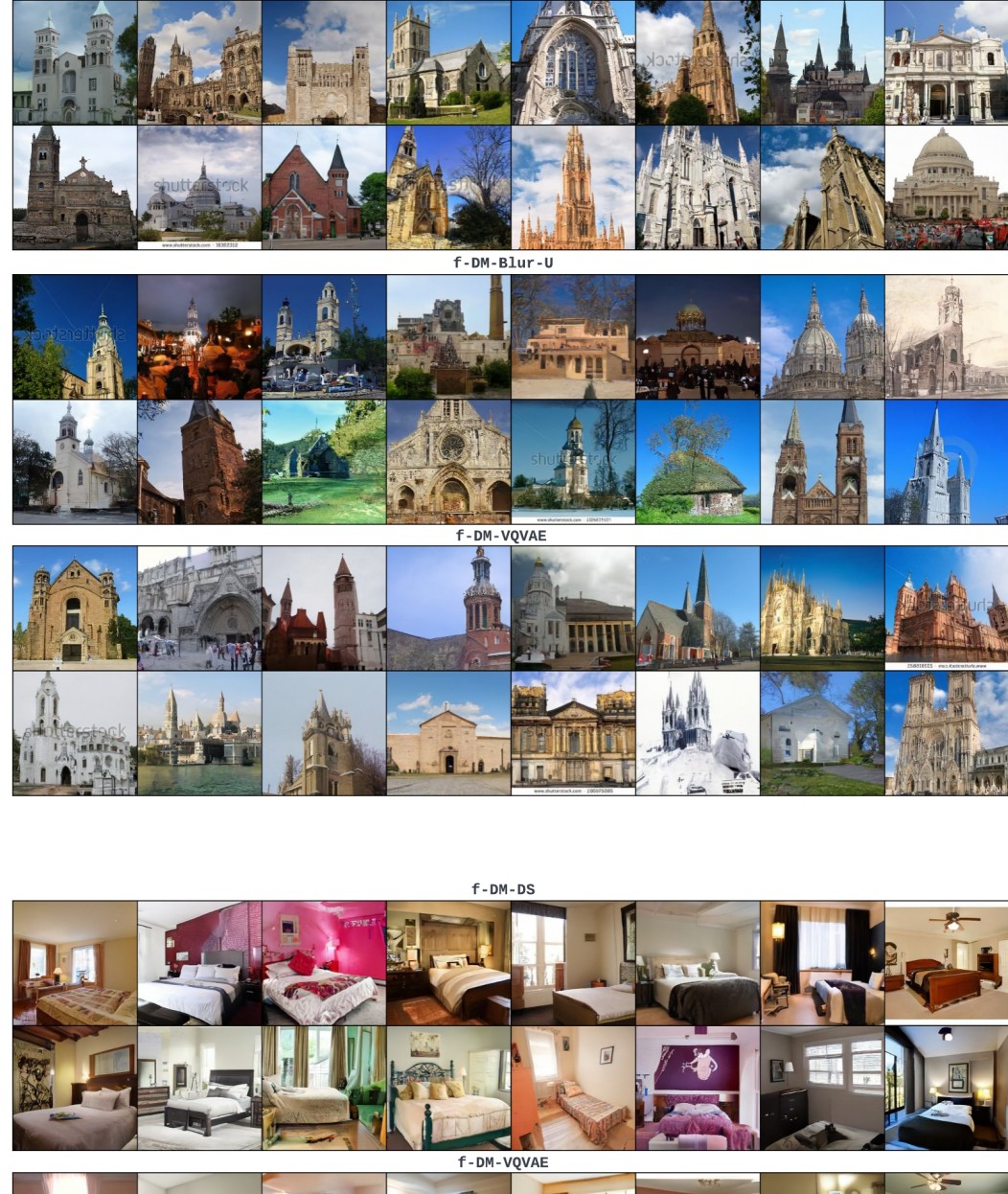

Figure 18: Random samples generated by $f$-DMs trained on LSUN-Church & -Bed $256 \times 256$.

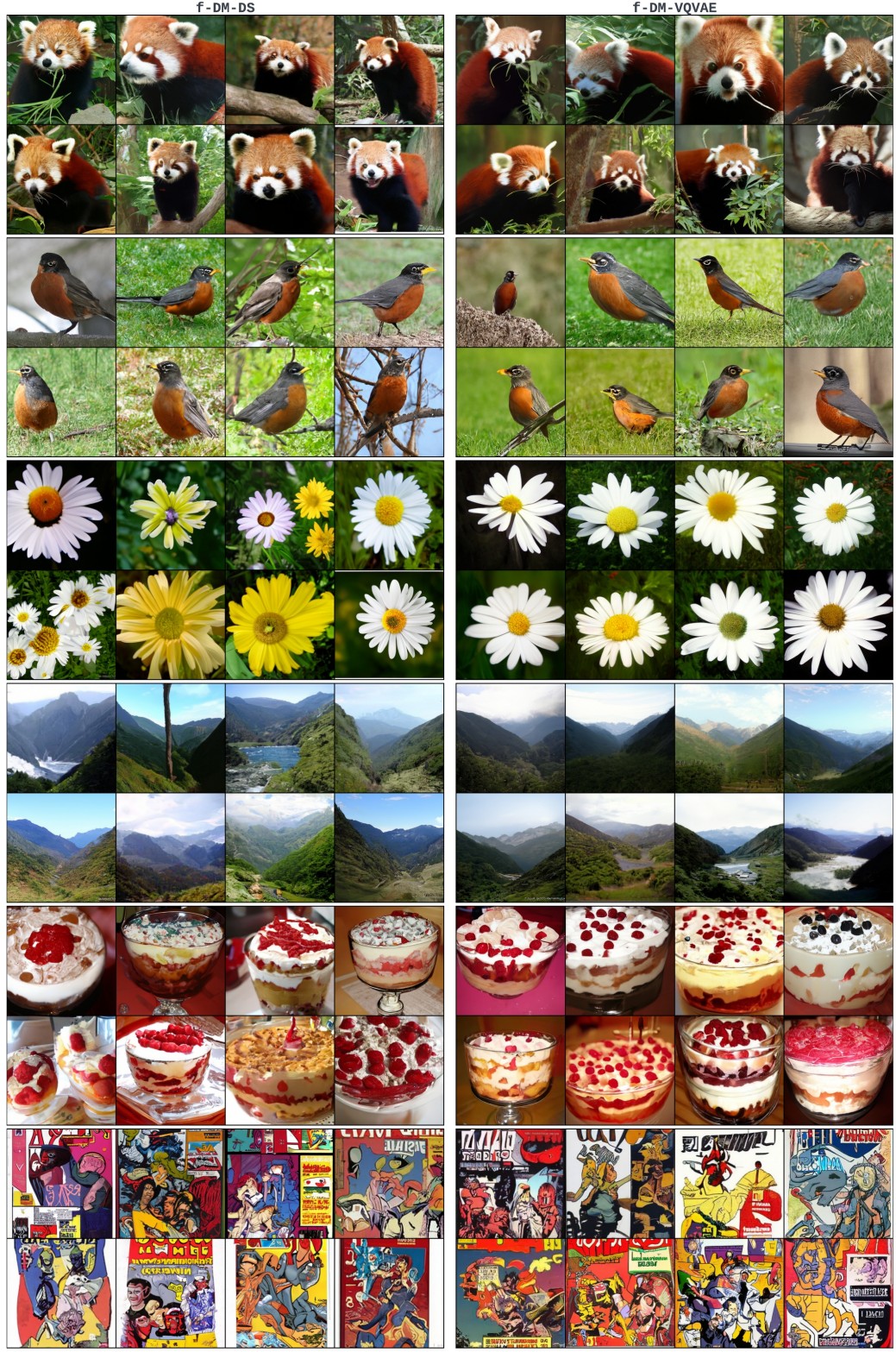

Figure 19: Random samples generated by $f$-DM-DS/VQVAE trained on ImageNet $256 \times 256$ with classifier-free guidance ($s = 3$). Classes from top to bottom: *red panda, robin, daisy, valley, trifle, comic book*.

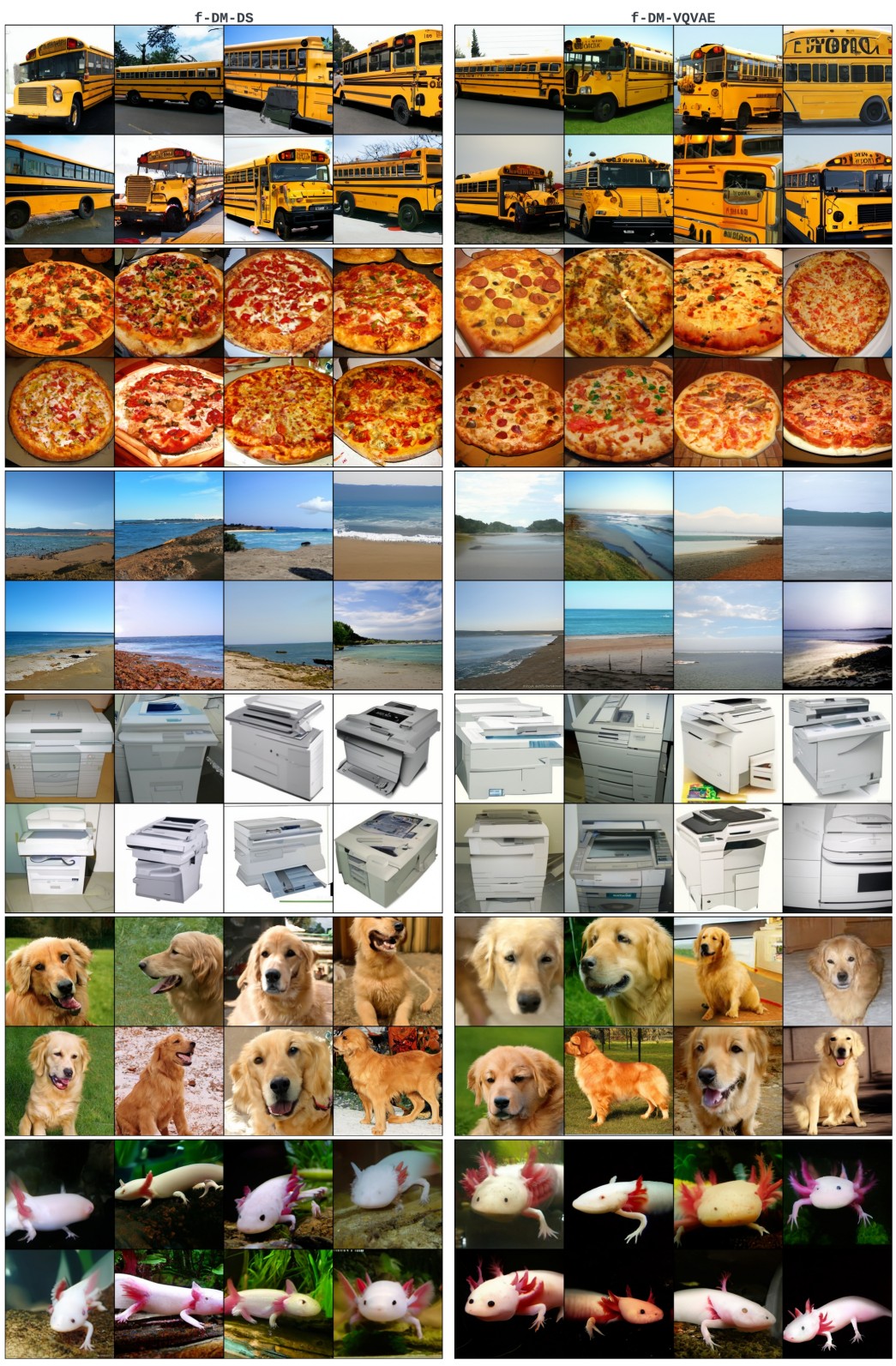

Figure 20: Random samples generated by $f$-DM-DS/VQVAE trained on ImageNet $256 \times 256$ with classifier-free guidance ($s = 3$). Classes from top to bottom: *school bus, pizza, seashore, photocopier, golden retriever, axolotl.*

