# OpenReview forum: "f-DM: A Multi-stage Diffusion Model via Progressive Signal Transformation"
_ICLR.cc/2023/Conference — ICLR 2023 poster_

### Official Review · Reviewer_Z8a2 · 2022-10-25

**Confidence:** 4
**Correctness:** 4
**Technical Novelty And Significance:** 3
**Empirical Novelty And Significance:** 4
**Recommendation:** 6

**Clarity, Quality, Novelty And Reproducibility:**

The paper is well written. The idea of using progressive signal transformation is also meaningful and novel.

**Strength And Weaknesses:**

Strength
- It is a solid idea that generalized denoising diffusions to other signal transformation tasks
- Experiments are performed a diverse set of generation tasks and datasets

Weakness
- The comparison with LDM, based on the experiments, is not conclusive. LDM (GAN) seems to be achieving the best FID, while Fig. 4 shows images with artifacts. Also, for more realistic and diverse datasets such as LSUN and ImageNet no comparison is performed with LDMs, while LDMs tend to achieve much better speed.

Questions and comments
- About f-DMs versus LDMs, it is not theoretically convincing if progressive transformation (such as down-sampling) can be more efficient. - At least from the experiments one can see that LDMs are much more efficient, and one needs to see the LDM trained with strong encoder and decoders on diverse datasets such as LSUN and ImageNet to make a conclusion.


**Summary Of The Paper:**

This submission deals with designing diffusion models by using progressive signal transformation. It proposes a generalized formulation of diffusion models, termed f-DM, with a modified sampling that is applied to image generation tasks with a range of signal transformations such as down-sampling, blurring, and learn compression based on pretrained VAEs. The reported experiments show better efficiency, quality and interpretation for image generation based on a few standard image datasets.


**Summary Of The Review:**

This paper deals with an important and timely problem. The proposed solution is also a solid generalization of denoising diffusions to general signal transformations and distortions. The reported experimental results also demonstrate significant FID improvement over some benchmarks. However, comparison with LDMs is not quite convincing and clear.

---

> ### Author Response · Authors · 2022-11-16
> **Response to Reviewer Z8a2**
>
> Thank you for your kind comments! We want to address the main confusion about the comparison with LDM as follows:
>
> > About f-DMs versus LDMs, it is not theoretically convincing if progressive transformation (such as down-sampling) can be more efficient. - At least from the experiments one can see that LDMs are much more efficient,
>
> **Re:** This paper does not claim better efficiency than LDM-like models. By choosing certain transformations (e.g., downsample, neural encoders) during diffusion, we can obtain better efficiency than vanilla diffusion models that act in the same pixel space. However, LDM directly works on a compact representation space by treating diffusion models as a black box algorithm. We will add a paragraph to clarify the efficiency comparison in the revised version.
>
> > The comparison with LDM, based on the experiments, is not conclusive. LDM (GAN) seems to be achieving the best FID, while Fig. 4 shows images with artifacts.
>
> **Re:** LDM is also not technically comparable.  An alternative view of f-DM vs LDM is that f-DM jointly trains the latent and pixel space generation in a unified way, whereas LDM is a two-stage approach where diffusion only happens in one space.
>
> Learning diffusion only in the latent space has both strengths and weaknesses. The performance of LDM heavily relies on the pretrained encoder-decoder. To achieve good reconstructions with a small bottleneck, the encoder-decoder has to be trained with adversarial training and perceptual objectives, which induces additional conceptual and practical complexity. As a result, artifacts caused by improper quantizations and adversarial loss are observed in generated samples (Appendix Figure 9).
>
> Moreover, FID is a biased metric that favors the fine details generated from adversarial training. Thus it might not be fair to compare LDM with f-DM with FID as they are using different objectives. Therefore, we also compare the visual quality of random samples. As another proof, we also compare the proposed method with LDM with a weaker VQ-VAE encoder/decoder, which in turn shows poorer results. By contrast, f-DM can easily adapt to any latent spaces and perform stably in both cases.

---

> ### Author Response · Authors · 2022-12-06
> **Any remaining concerns?**
>
> We again thank the reviewer for their time and their reviews of our submission! We hope our rebuttal successfully addresses the reviewer's suggestions.
>
> We are happy to further address any remaining concerns not covered in the rebuttal that the reviewer may have before making their final assessment.

---

### Official Review · Reviewer_FRYF · 2022-10-27

**Confidence:** 4
**Correctness:** 4
**Technical Novelty And Significance:** 4
**Empirical Novelty And Significance:** 3
**Recommendation:** 8

**Clarity, Quality, Novelty And Reproducibility:**

Clarity: Could be improved - see weaknesses above

Quality: See strengths/weaknesses.

Novelty: Novel AFAIK

Reproducibility: Contains all required details as far as I can tell. The authors also commit to releasing code.

**Strength And Weaknesses:**

Strengths:
- At a high level, I think this is an elegant method to integrate transformations into a diffusion process. This could be of considerable interest to the ICLR community.
- Well thougt-out design choices including interpolating between $\hat{x}_k$ and $\hat{x}_k$ at each stage between transformations (as described in Equation 4), parameterisation of the network to predict both $\epsilon$ and $\delta$, and definition of $q(z_t|x)$.
- Good experimental results.


Weaknesses:
 - Slightly confusing presentation - reading Section 3.1 was reasonably difficult. I think I understand it after spending considerable time on it, but suspect that it could be made clearer with more explanation (or perhaps a diagram/expanded version of Fig 2b) of the relationships between each $x$, $\hat{x}$, $z_t$, etc.
- In Equation 7, $\Omega$ is said to be the "minimal interested patch", with no further explanation. And the "SIGNAL" and "NOISE" functions in Eq. 7 are not explained at all. The conclusions reached about setting $\alpha_{\tau}$, $\alpha_{\tau^-}$, $\sigma_{\tau}$, and $\sigma_{\tau^-}$ seem reasonable, but their relationship to Eq. 7 needs more explanation.

**Summary Of The Paper:**

The authors propose a diffusion model which incorporates functions other than the addition of Gaussian noise in the forward process. Concretely, they show results with downsampling functions, blurring, and a learned neural encoding.

**Summary Of The Review:**

The proposed method seems to be well-thought out and of interest to the ICLR community. I recommend acceptance, although the clarity/presentation could still be improved.

---

> ### Author Response · Authors · 2022-11-16
> **Response to Reviewer FRYF**
>
> Thank you for your kind comments! The following are our responses to your concerns:
>
> > Slightly confusing presentation - reading Section 3.1 was reasonably difficult. I think I understand it after spending considerable time on it, but suspect that it could be made clearer with more explanation (or perhaps a diagram/expanded version of Fig 2b) of the relationships between each $x$, $\hat{x}$, $z_t$, etc.
>
> **Re:** We agree that the original notions may cause some confusion. We will rewrite Section 3.1 in the revised version to make it clearer to show our method and notations. We also include Figure 4 in the updated paper, which offers an example of a training step to clearly explain the relationship between $x^k$, $x^{k+1}$, $\hat{x}^k$, $x_t$, and $z_t$.
>
> > Eq. 7 needs more explanation.
>
> **Re:** Thanks for pointing out the unclear part in Eq 7, and we will include more explanations for the terms used in the revised version. We also have explained the main intuition behind “resolution-agnostic” SNR and rescaling in Appendix A4.

---

> ### Author Response · Authors · 2022-12-06
> **Addressing remaining concerns**
>
> We again thank the reviewer for their time and their reviews of our submission!
> We hope our rebuttal successfully addresses the reviewer's suggestions.
>
> We are happy to further address any remaining concerns not covered in the rebuttal that the reviewer may have before making their final assessment.

---

### Official Review · Reviewer_C5sD · 2022-11-03

**Confidence:** 4
**Correctness:** 3
**Technical Novelty And Significance:** 2
**Empirical Novelty And Significance:** 2
**Recommendation:** 5

**Clarity, Quality, Novelty And Reproducibility:**

The paper is well written. However, it has limited technical novelty (see weakness).

**Strength And Weaknesses:**

Strength:
1. The paper is well-written with thorough experimental results.
2. This paper considers a very interesting direction of diffusion models by incorporating progressive signal transformation.

Weakness:
1. Limited novelty: Similar ideas (e.g., incorporating progressive signal transformation into generative modeling) have be extensively explored for other generative models including GANs [1], normalizing flows [2,3], and VAEs. For the three major transformations considered in the paper (down-sample, gaussian blur and VQ-VAE/VQ-GAN), there are already existing work on diffusion models using each of the transformation as mentioned in the paper. Specifically, for transformations learned by neural networks, diffusion models trained on a latent space fit into this category: there have been various works in this direction and it is not a new idea. For instance, besides LDM, [4,5,6] also perform diffusion on data transformed with an encoder. In some sense, the contribution of this paper seems to unify these approaches into one framework by calling them "diffusion models learned with progressive signal transformation".

2.  Limited theoretical insights: the contribution of this paper is mainly on the empirical side. There are not enough theoretical insights or guarantee.

3. It seems that the performance (e.g., sample quality/speed) depends heavily on the transformation $f$ and the stage scheduling. More discussion should be provided on the selection of the optimal tranformations. Although the transformation are claimed to be any transformation, it remains unclear how effective/useful the other transformations are besides the ones used in the paper  (down-sample, Gaussian blur and transformations learned from a neural network). However, there are already existing work using these transformations.

4. In the experiment, the reimplemented Cascaded DM (figure 4) seems to have much worse performance than the ones reported in the original paper. It is unclear how trustworthy the results are.

5. For comparison with DDPM with fewer steps (DDPM 1/2), the authors should also consider comparing with DDIM 1/2 [7] in Table 1.

6. How expensive it is to find the stage scheduling used in the experiments? Is there any theoretical connection to ODE/SDE?

[1] Progressive Growing of GANs for Improved Quality, Stability, and Variation: https://arxiv.org/abs/1710.10196

[2] Wavelet Flow: Fast Training of High Resolution Normalizing Flows: https://arxiv.org/abs/2010.13821

[3] Improving Continuous Normalizing Flows
using a Multi-Resolution Framework: https://arxiv.org/abs/2106.08462

[4] D2C: Diffusion-Denoising Models for Few-shot Conditional Generation: https://arxiv.org/abs/2106.06819

[5] Diffusion-LM Improves Controllable Text Generation: https://arxiv.org/abs/2205.14217

[6] Symbolic Music Generation with Diffusion Models: https://arxiv.org/abs/2103.16091

[7] Denoising Diffusion Implicit Models: https://arxiv.org/abs/2010.02502


**Summary Of The Paper:**

This work proposes to train diffusion models via multi-stage progressive signal transformations. Compared to traditional diffusion models, the signal transformation here is more general and can be any transformation from coarse to fine. The authors focus on three main transformations (down-sample, Gaussian blur and transformations learned from a neural network) in the experiments, and demonstrate the effectiveness of the approach.

**Summary Of The Review:**

This work considers learning diffusion models on data processed with a sequence of signal transformations. Similar ideas have been explored in other generative models, including normalizing flows, GANs and VAEs. It is almost straightforward to apply similar ideas to diffusion models. Although the proposed approach is claimed to work for any transformation, it remains unclear how effective it would be for random transformations besides the ones the authors performed experiments on. At the same time, there are already works that learn diffusion models on down-sampled images or latent spaces. Given these related works, the novelty of this work is questionable.

---

> ### Author Response · Authors · 2022-11-16
> **Response to Reviewer C5sD (1/N)**
>
> Thank you for your kind comments! The following are our responses to your concerns:
>
> ### Novelty
>
> > Similar ideas (e.g., incorporating progressive signal transformation into generative modeling) have be extensively explored for other generative models including GANs [1], normalizing flows [2,3], and VAEs.
>
> **Re:** Thanks for pointing out this connection; we are fully aware of it. Our paper is, in fact, directly motivated by these approaches, where signals are transformed in different spaces during the generation with useful properties (e.g., multi-scale abstraction). However, the standard DMs are incompatible with these transformations, and we have not seen any variant of DMs that organically incorporate them in a single framework. Therefore, while our motivation is straightforward, our contribution to implementing such an idea is novel and non-trivial (**explained in detail below**). That being said, we are happy to include all the mentioned methods in our related work section and include proper discussions.
>
> > For the three major transformations considered in the paper (down-sample, gaussian blur, and VQ-VAE/VQ-GAN), there are already existing work on diffusion models using each of the transformation as mentioned in the paper. Specifically, for transformations learned by neural networks, diffusion models trained on a latent space fit into this category: there have been various works in this direction and it is not a new idea. For instance, besides LDM, [4,5,6] also perform diffusion on data transformed with an encoder.
>
> **Re:** We would like to emphasize that f-DM is **significantly different from** learning a DM on a “transformed space” (e.g., low-resolution images and latent space from an encoder)  in that our data transformation is progressive. What these references have in common is that they treat DM as a black-box generative model and focus on transforming the input representation. By contrast, we directly generalize the forward and backward process of DMs where progressive signal transformation is coupled with diffusion/denoising. We show that this progressive transformation makes the diffusion process more reversible as the data corruption is more incremental. This generally improves the generation quality. We selected "downsampling," "blurring," and "learned latent spaces" as three representatives to show the effectiveness of our methods, but f-DM is a general framework that applies to a wide set of transformations.
>
>
> > In some sense, the contribution of this paper seems to unify these approaches into one framework by calling them "diffusion models learned with progressive signal transformation".
>
> **Re:**  On the technical side, enabling diffusion via progressive transformations is more than simply unifying existing approaches. We identified two major challenges: (1) handling the change of representation space and its corresponding information loss and (2) the need to rescale the noise scheduling in the diffusion process when there is a dimension change. For (1), as correctly *pointed out by Reviewer FRYF*, we specifically designed the interpolation formulation between consecutive transformations and the double prediction parameterization; for (2), a resolution-agonistic SNR was proposed to guide the noise-rescaling across the transformation boundary. These technical innovations are not discovered by any prior works, and we found them essential to make f-DM match or even beat standard DDPM.
>
> Based on the points above, we believe the novelty of this work is justified.
>
> ---
>
> ### Theoretical insight & connection to SDE/ODE
>
> > Limited theoretical insights: the contribution of this paper is mainly on the empirical side. There are not enough theoretical insights or guarantee.
>
> > Is there any theoretical connection to ODE/SDE?
>
> **Re:**  We would like to address the theoretical connection as follows:
>
> * First, f-DM is a rigorous likelihood-based model trained by a variant of the MLE objective. It also strictly generalizes DPMs, which implies that it shares the same theoretical insights as standard DPMs.
>
> * Second, f-DM, its whole, cannot be directly described with an SDE/ODE language. However, we would also like to argue that the SDE/ODE formulations have their limitations. In particular, they assume fixed dimensionality and the same space throughout the diffusion process, which induces high computational costs and little design flexibility. f-DM is different from SED/ODE by relaxing these constraints, which we believe is an advantage rather than a drawback.
>
> * Lastly, each step of f-DM can still be described by an SDE/ODE and benefit from the same continuous time parameterization and interpretation. One can view our methods as learning a group of SDEs jointly, each of which may have different dimensionality. The forward diffusion starts from a high dimensional SDE and jumps to lower dimensions when the equation evolves. We will have more concrete descriptions in the next version.

---

> ### Author Response · Authors · 2022-11-16
> **Response to Reviewer C5sD (2/N)**
>
> ### Performance depending on transformations & generality to transformations
>
> > It seems that the performance (e.g., sample quality/speed) depends heavily on the transformation f and the stage scheduling. More discussion should be provided on the selection of the optimal tranformations. Although the transformation are claimed to be any transformation, it remains unclear how effective/useful the other transformations are besides the ones used in the paper (down-sample, Gaussian blur and transformations learned from a neural network). However, there are already existing work using these transformations.
>
> **Re:**
> * First, as shown in Table 1 and the generation samples in Appendix E, the performance of f-DM is relatively robust wrt the choice of transformations. In particular, most models achieve similar FID scores (difference less than 2) and visual quality, which matches or even beats the standard DDPM without additional transformations.
>
> * Second, the fact that our method performs stably on two learned transformations (VQVAE/VQGAN encoders), indicates that f-DM is general and compatible with arbitrary linear or non-linear transformations (because neural nets are universal approximators). People can easily incorporate their inductive bias into transformations. For example, we can design wavelet-based transformations to diffuse in the frequency domain.
>
> * That being said, we believe that defining “optimal” transformations is also an open question. We believe that f-DM enables the possibility of exploring a wide space of transformations in a single framework, and we look forward to seeing future works in this direction.
>
> ---
>
> ### Comparison to Cascaded DM
>
> > In the experiment, the reimplemented Cascaded DM (figure 4) seems to have much worse performance than the ones reported in the original paper. It is unclear how trustworthy the results are.
>
> **Re:** Cascaded DM is closely related to f-DM with downsampling as the transformation. We then conducted comparisons with the same code base and hyperparameters. Our cascaded DM differs from the original paper in two notable ways:
>
> * (1) We apply more levels of downsampling (5 for 256x256 images) compared to existing papers (generally 1 to 3 levels). More levels typically mean more challenges for cascaded DM because of the error propagation.
>
> * (2) We did not apply “noise augmentation” when training upsampling models in cascaded DM, a heuristic designed to reduce the domain gaps between training and testing (See Sec 3 of the Cascaded Diffusion Model paper [1] for detailed discussions). It is not clear how to tune the augmented noise for each dataset. f-DM, on the other hand, provides a principled guideline on rescaling the noise schedule according to a reference one.
>
> We believe that our comparisons are valid and fair. The fact that our Cascaded DM yields worse performance than the default settings suggests that Cascaded DM is sensitive to the hyperparameters and not easily reproducible.
>
> The revised version will highlight the similarity and differences between cascaded models.
>
> ---
>
> ### Comparison with DDIM
>
> > For comparison with DDPM with fewer steps (DDPM 1/2), the authors should also consider comparing with DDIM 1/2 [7] in Table 1.
>
> **Re:** To achieve a fair comparison between different models, the experimental quantitive comparison is based on DDPM sampling, which is known to have better sampling quality but is relatively more sensitive to the number of iteration steps due to the sampling noise. However, we can still see clear degradation on FID when using half of the steps (250 → 125), even with DDIM sampling. We will include the exact numbers of DDIM comparisons of each dataset in the revised version.
>
> ---
>
> ### Concerns about finding stage scheduling
>
> > How expensive it is to find the stage scheduling used in the experiments?
>
> **Re:** This paper does not focus on searching for the optimal stage scheduling for each transformation. Like the noise schedule, we treat the stage schedule as another hyper-parameter, and we tested two types of scheduling (linear and cosine) and kept the same noise schedule for all experiments. Based on our observation, we found the results are stable, but transformations have slightly different preferences (shown in the Ablation study in Table 2). We conjectured the optimal stage schedule might correlate to how fast the information loses during diffusion, and we leave this as one of our future works.
>
> ---
>
> [1] Ho, Jonathan, et al. "Cascaded Diffusion Models for High Fidelity Image Generation." J. Mach. Learn. Res. 23 (2022): 47-1.

---

> ### Author Response · Authors · 2022-12-06
> **Any remaining concerns?**
>
> We again thank the reviewer for their time and their reviews of our submission!
> We hope our rebuttal successfully addresses the reviewer's suggestions.
>
> We are happy to further address any remaining concerns not covered in the rebuttal that the reviewer may have before making their final assessment.

---

### Author Response · Authors · 2022-11-19
**General response on the updated version**

Dear reviewers,

Thank you for your insightful comments and suggestions! We have updated the rebuttal revision of this paper, which mainly covers the following points:

* Adding additional comparison with DDIM in Appendix E1 as requested by C5sD.

* Adding Figure 4 to explain the relationships between variables ($x_t$, $x^k$, etc)

* Adding more explanation in Eq 7 and other equations.

* General content updates in the Appendix for additional information (such as model parameters, dataset, and transformation details, details on stage schedule (Figure 8), and details about DDIM sampling with f-DM (Figure 9)) to support better reproducibility

---

### Decision · Program_Chairs · 2023-01-20

**Decision:**

Accept: poster

**Justification For Why Not Higher Score:**

The contribution could be clearer formulated.

**Justification For Why Not Lower Score:**

Good solid contribution within a red-hot area.

**Metareview: Summary, Strengths And Weaknesses:**

The paper proposes different types of learned and fixed encoders for DDPMs and tests the proposed schemes against other models for a range of benchmarks.

All reviewers appreciate the contribution and vote for acceptance.

**Note From Pc:**

if the above contains the word "oral" or "spotlight" please see: "oral" presentation means -> notable-top-5% and "spotlight" means -> notable-top-25%. As stated in our emails, we are disassociating presentation type from AC recommendations